# Online Selective Classification with Limited Feedback

**Aditya Gangrade**
Boston University
gangrade@bu.edu

**Anil Kag**
Boston University
anilkag@bu.edu

**Ashok Cutkosky**
Boston University
ashok@cutkosky.com

**Venkatesh Saligrama**
Boston University
srv@bu.edu

## Abstract

Motivated by applications to resource-limited and safety-critical domains, we study selective classification in the online learning model, wherein a predictor may abstain from classifying an instance. For example, this may model an adaptive decision to invoke more resources on this instance. Two salient aspects of the setting we consider are that the data may be non-realisable, due to which abstention may be a valid long-term action, and that feedback is only received when the learner abstains, which models the fact that reliable labels are only available when the resource intensive processing is invoked.

Within this framework, we explore strategies that make few mistakes, while not abstaining too many times more than the best-in-hindsight error-free classifier from a given class. That is, the one that makes no mistakes, while abstaining the fewest number of times. We construct simple versioning-based schemes for any $\mu \in (0, 1]$, that make most $T^\mu$ mistakes while incurring $\tilde{O}(T^{1-\mu})$ excess abstention against adaptive adversaries. We further show that this dependence on $T$ is tight, and provide illustrative experiments on realistic datasets.

## 1  Introduction

Consider a low-power or battery-limited edge device, such as a sensor or a smart-speaker that receives a stream of classification requests. Due to the resource limitations, such a device cannot implement modern models that are needed for accurate decisions. Instead the device has access (e.g. via an internet connection) to an accurate but resource-intensive model implemented on a cloud server, and may send queries to the cloud server in order to retain accuracy. Of course, this incurs costs such as latency and battery drain due to communication. The ideal operation of such a device should thus be to learn a rule that classifies 'easy' instances locally, while sending harder ones to the cloud, thus maintaining accuracy whilst minimising the net resource consumption [Xu+14; NS17].

Selective classification [Cho57; Cho70] is a classical paradigm of relevance to such settings. The setup allows a predictor to abstain from classifying some instances (without incurring a mistake). This abstention models adaptive decisions to invoke more resource-intensive methods on subtle cases, like in the above example. The solution concept is relevant widely - for instance, it is relevant to adaptively recommending further (and costly) tests rather than offering a diagnosis in a medical scenario, or to recommending a human review instead of an alarm-or-not decision in security contexts. Two aspects of such settings are of particular interest to us. Firstly, the cheaper methods are typically not sufficient to realise the true labels, due to which abstention may be a long-term necessity. Secondly, a-priori reliable labels can only be obtained by invoking the resource intensive option, and thus feedback on whether a non-abstaining decision was correct is unavailable.

35th Conference on Neural Information Processing Systems (NeurIPS 2021).

We propose online selective classification, with an emphasis on ensuring very few mistakes, to account for the need for very accurate decisions. Concretely, an adversary sequentially produces contexts and labels $(X_t, Y_t)$, and the learner uses the $X_t$s to produce a decision $\widehat{Y}_t$ that may either be one of $K$ classes, or an abstention, which we represent as $\perp$. Feedback in the form of $Y_t$ is provided if and only if $\widehat{Y}_t = \perp$, and the learner incurs a mistake if $\widehat{Y}_t$ was non-abstaining and did not equal $Y_t$.

With the emphasis on controlling the total number of mistakes, we study regrets achievable when compared to the behaviour of the best-in-hindsight error-free selective classifier from a given class - that is, one that makes no mistakes, while abstaining the fewest number of times. Notice that our situation is non-realisable, and therefore this competitor may abstain in the long-run. The two metrics of importance here are the number of mistakes the learner makes, and its excess abstention over this competitor. An effective learner must control both abstention and mistakes, and it is not enough to make one small, e.g. a learner that makes a lot of mistakes but incurs a very negative excess abstention is no good. This *simultaneous* control of two regrets raises particular challenges.

We construct a simple scheme that, when competing against finite classes, simultaneously guarantees $O(T^\mu)$ mistakes and $O(T^{1-\mu})$ excess abstentions against adaptive adversaries (for any $\mu \in [0, 1]$), and show that these rates are Pareto-tight [OR94]. We further show that against stochastic adversaries, the same rates can be attained with improved dependence of the regret bounds on the size of the class, and we also describe schemes that enjoy similar improvements against adaptive adversaries, but at the cost of the $T$-dependence of the regret bounds. The main schemes randomly abstain at a given rate in order to gain information, and otherwise play $\widehat{Y}_t$ consistent with the 'version space' of classifiers that have not been observed to make mistakes. For the adversarial case, the analysis of the scheme relies on a new 'adversarial uniform law of large numbers'(ALLN) to argue that such methods cannot incur too many mistakes. This ALLN uses a self-normalised martingale concentration bound, and further yields an adaptive continuous approximation guarantee for the Bernoulli-sampling sketch in the sense of Ben-Eliezer & Yogev [BY20; Alo+21]. The theoretical exploration is complemented by illustrative experiments that implement our scheme on two benchmark datasets.

## 1.1 Related Work

Selective classification has been well studied in the batch setting, and many theoretical and methodological results have appeared [e.g. HW06; BW08; EW10; WE11; KKM12; CDM16; Lei14; GE19; GKS21]. These batch results do not have strong implications for the online setting.

Cortes et al. have studied selective classification in the online setting [Cor+18], but with two differences from our setting. Firstly, rather than individually controlling mistakes and abstentions, the regret is defined according to the Chow loss, which adds up the number of mistakes and $c$ times the number of abstentions, where $c$ is a fixed cost parameter. Secondly (and more importantly) it is assumed that feedback is provided only when the learner *does not* abstain, rather than only when it does. This difference arises from the underlying situations being modelled - Cortes et al. view the abstention as a decision given to a user in which case no feedback is forthcoming, while we view it as a decision to invoke further processing. Both of the scenarios are reasonable, and so both of these explorations are valid, however it is unclear what implications one set of results have for the other.

A similar decision and feedback model as ours was proposed by Li et al. in the 'knows what it knows' (KWIK) framework [LLWS11]. The KWIK model, however, fundamentally views abstentions as a short term action, typically arguing that only a finite number of these are made. This is viable since Li et al. study this model in an essentially realisable setting, wherein the optimal labels are known to be essentially realised by a given class - notice that in such a case, a single abstention at an instance $x$ determines what value should be played there in the long run. Our interest however lies in the situation where this data cannot be represented in such a way, and such strategies are not viable since the labels may be noisy. Our work thus generalises the KWIK setting to non-realisable data, and to situations wherein abstention is a valid long-term action, as motivated in the introduction, by studying behaviour against competitors that may abstain.[1]

While Szita and Szepesvári have extended the KWIK formulation to the agnostic case in a regression setting [SS11], this work also focuses of limiting the number of abstentions to be finite rather than

---

[1]The KWIK model also bears other significant differences. It posits an input parameter $\varepsilon$, and requires that the learner either abstains, or produces an $\varepsilon$-accurate response. A notion of competitor is not invoked, and rather than studying regret, the number of abstentions needed to achieve this $\varepsilon$-accuracy is studied.

long-run abstentions. Concretely it is assumed that $Y_t = g(X_t) + \text{noise}$, for some function $g$, and the learner knows a class $\mathcal{H}$, and a bound $\Delta$ such some $h \in \mathcal{H}$ is $\Delta$-close to $g$ (in an appropriate norm). Using the knowledge of $\Delta$, they describe schemes that have limited abstention, but at the cost of mistakes, by producing responses $\hat{Y}_t$ that are up to $(2 + o(1))\Delta$ separated from $Y_t$. In contrast, in our formulation, contexts $X_t$ for which no function in $\mathcal{H}$ can represent the ground truth $g$ well would always be abstained upon. In addition to this work, trade-offs between mistakes and abstentions in a relaxed version of the KWIK framework have been considered [ZC16; SZB10; DZ13], and in particular the agnostic case has been explored by Zhang and Chaudhuri [ZC16], but unlike our situation this relaxed KWIK model requires full-feedback to be available whether or not the learner abstains. Neu and Zhivotovskiy [NZ20] also work in this relaxed model, and show that when comparing the standard loss of a *non-abstaining* classifier against the Chow loss of an abstaining learner, regrets independent of time can be obtained.

Due to the limited feedback, our setting is related to partial-monitoring [LS20, Ch. 37]. Viewing actions as choices over functions, our setting has feedback graphs [MS11] that connect abstaining actions to every other action and themselves. The novelty with respect to partial-monitoring arises from the fact that we individually control two notions of losses, rather than a single one. It's unclear how to apply the generic partial-monitoring setup to this situation - indeed, naïvely, our game is only weakly observable in the sense of Alon et al.[ACDK15], and one would expect $\Omega(T^{2/3})$ regrets, while we can control both mistakes and excess abstention to $\tilde{O}(\sqrt{T})$. A limited feedback setting where two 'losses' *are* individually controlled is label-efficient prediction [CLS05], where a learner must query in order to get feedback. However, in our setting, abstentions are both a way to gather feedback, and also necessary to prevent mistakes. That is, our competitor may abstain regularly, but makes few mistakes, while in this prior work the competitor does not abstain, but may make many mistakes. The resulting scenario is both qualitatively and quantitatively distinct, e.g. in label-efficient prediction, the smallest symmetric rate of number of queries and excess mistakes is again $\Theta(T^{2/3})$.

## 2 Setting, and Problem Formulation

**Setup** Let $\mathcal{X}$ be a feature space, $\mathcal{Y}$ a finite set of labels, and $\mathcal{F}$ a finite class of selective classifiers, which are $\mathcal{Y} \cup \{\perp\}$ valued. For simplicity, we assume that $\mathcal{F}$ contains the all abstaining classifier (i.e. the function $f_\perp$ such that $\forall x, f_\perp(x) = \perp$). We will denote $|\mathcal{F}| = N$. The setting may be described as a game between a learner and an adversary (or more prosaically, a data generating mechanism) proceeding in $T$ rounds. Also for simplicity, we will assume that $T$ is known to both the learner and the adversary in advance. The objects in this game are the context process, $X_t \in \mathcal{X}$, the label process $Y_t \in \mathcal{Y}$, the action process $\hat{Y}_t \in \mathcal{Y} \cup \{\perp\}$ and the feedback process $Z_t \in \mathcal{Y} \cup \{*\}$, where $* \notin \mathcal{Y}$ is a trivial symbol. The information sets of the adversary and learner up to the $t$th round are respectively $\mathscr{H}^{\mathfrak{A}}_{t-1} := \{(X_s, Y_s, \hat{Y}_s) : s < t\}$, and $\mathscr{H}^{\mathfrak{L}}_{t-1} := \{(X_s, \hat{Y}_s, Z_s) : s < t\}$.

**The Game** For each round $t \in [1 : T]$, the adversary produces a context and a label $(X_t, Y_t)$ on the basis its history $\mathscr{H}^{\mathfrak{A}}_{t-1}$. The learner observes only the context, $X_t$, and on the basis of this and its history $\mathscr{H}^{\mathfrak{L}}_{t-1}$, produces an action $\hat{Y}_t$. We will say that this action is an abstention if $\hat{Y}_t = \perp$, and that it is a prediction otherwise. If the action was an abstention, set $Z_t = Y_t$, and otherwise to $*$. The learner then observes $Z_t$, and the round concludes. Notice that since $Z_t$ is a deterministic function of $Y_t$ and $\hat{Y}_t$, and since the adversary observes both, $\mathscr{H}^{\mathfrak{L}}_{t-1}$ can be determinstically generated from $\mathscr{H}^{\mathfrak{A}}_{t-1}$. Due to the same reason, $\hat{Y}_t$ and $Y_t$ are conditionally independent given $(X_t, \mathscr{H}^{\mathfrak{A}}_{t-1})$.

**Adversaries** are characterised by a sequence of conditional laws on $(X_t, Y_t)$ given $\mathscr{H}^{\mathfrak{A}}_{t-1}$ (and $T, \mathcal{F}$). In the following we will explicitly consider two classes of such laws:
• Stochastic Adversary: $(X_t, Y_t)$ are drawn according to a fixed law, $P$, unknown to the learner, independently of $\mathscr{H}^{\mathfrak{A}}_{t-1}$.
• Adaptive Adversary: $(X_t, Y_t)$ are arbitrary random variables with $\mathscr{H}^{\mathfrak{A}}_{t-1}$-measurable laws.
We will denote a generic class of adversaries as $\mathscr{C}$.

**Performance Metrics** The two principal quantities of interest are the number of mistakes made by the learner, and the number of times it has abstained. We will denote these as

$$M_T := \sum_{t \leq T} \mathbb{1}\{\hat{Y}_t \notin \{\perp, Y_t\}\}, \quad \text{and} \quad A_T := \sum_{t \leq T} \mathbb{1}\{\hat{Y}_t = \perp\}.$$

As previously discussed, the performance of a learner is measured in terms of regret with respect to the best-in-hindsight abstaining classifier from $\mathcal{F}$ that makes no mistakes, that is

$$f^* \in \arg\min_{f \in \mathcal{F}} \sum_{t \leq T} \mathbb{1}\{f(X_t) = \bot\} \quad \text{s.t.} \quad \sum_{t \leq T} \mathbb{1}\{f(X_t) \notin \{\bot, Y_t\}\} = 0.$$

Note that such an $f^*$ is always realised, since the class is finite, and since it contains the all abstaining classifier. Let $A_T^* := \sum_{t \leq T} \mathbb{1}\{f^*(X_t) = \bot\}$ denote the value of the minimum above. The principal metrics of interest to us are the *abstention regret* $A_T - A_T^*$, and the *total mistakes* $M_T$.

**Solution Concept** The two performance metrics naturally involve a tradeoff - for instance, making some mistakes may allow a learner to drastically reduce its abstention regret to the point that it is negative. We pursue the trade-off between the worst possible behaviour of either regret.

**Definition** (Regret Achievability) *For functions $\varphi, \psi : \mathbb{N}^2 \to \mathbb{R}$, we say that expected regret bounds of $(\varphi, \psi)$ are achievable against a class of adversaries $\mathcal{C}$ if there exists a learner such that for every adversary in $\mathcal{C}$, $\mathbb{E}[A_T - A_T^*] \leq \varphi(T, N)$ and $\mathbb{E}[M_T] \leq \psi(T, N)$.*

As is common, we are interested in the growth rates of achievable bounds with $T$. We thus define

**Definition** (Achievable rates) *we say that asymptotic expected-regret rates of $(\alpha, \mu) \in [0, 1]^2$ are achievable against a class of adversaries $\mathcal{C}$ if an expected regret bound of $(\varphi, \psi)$ can be achieved against it for functions $\varphi, \psi$, said to be witnesses for the rate, such that*

$$\limsup_{T \to \infty} \frac{\log \varphi(T, N)}{\log T} \leq \alpha \quad \text{and} \quad \limsup_{T \to \infty} \frac{\log \psi(T, N)}{\log T} \leq \mu.$$

Notice that if $(\alpha, \mu)$ is an achievable rate, so is $(\alpha', \delta')$ for $\alpha' \geq \alpha, \delta' \geq \delta$. As a result, the lower boundary of the set of achievable rates is well defined, and we will refer to this as the *Pareto frontier of achievable rates*. This is equivalently characterised by the function $\underline{\alpha}(\mu) := \inf\{\alpha : (\alpha, \mu) \text{ is an achievable rate}\}$. This is well defined since $\forall \mu, (1, \mu)$ is achievable by always abstaining.

## 3 The Adversarial Case

We begin with the adversarial case. The scheme, called the 'versioned uniform explorer' (VUE) is described below, and we discuss both the motivation of the scheme, and its analysis.

The main idea underlying VUE is that any function $f$ that is observed to make a mistake on an instance $X_t$ (due to the learner abstaining on this instance) can be removed from future consideration, since we are only trying to match the behaviour of the competitor $f^*$, and clearly $f \neq f^*$ as it has made a mistake. This motivates setting up a 'version space,'

$$\mathcal{V}_t := \left\{ f : \sum_{s < t} \mathbb{1}\{Z_s \neq *, f(X_s) \notin \{\bot, Y_s\}\} = 0 \right\},$$

the set of functions that are consistent with the observations made up to time $t$. Notice that $f^* \in \mathcal{V}_t$ for all $t$. Given $\mathcal{V}_t$, we can restrict to playing an action in the set $\widehat{\mathcal{Y}}_t := \{f(X_t) : f \in \mathcal{V}_t\}$ - $f^*(X_t)$ lies in this set, and thus any action outside of it can be eliminated. Of course, if $\widehat{\mathcal{Y}}_t$ is a singleton, then it contains $f^*(X_t)$, and we can just play it.

---
**Algorithm 1** VUE

1: **Inputs**: $\mathcal{F}$, Exploration rate $p$.
2: **Initialise**: $\mathcal{V}_1 \leftarrow \mathcal{F}$.
3: **for** $t \in [1 : T]$ **do**
4:     $\widehat{\mathcal{Y}}_t \leftarrow \{f(X_t) : f \in \mathcal{V}_t\}$.
5:     **if** $|\widehat{\mathcal{Y}}_t| = 1$ **then**
6:         $\widehat{Y}_t \leftarrow f(X_t)$ for any $f \in \mathcal{V}_t$.
7:         $\mathcal{V}_{t+1} \leftarrow \mathcal{V}_t$.
8:     **else**
9:         Sample $C_t \sim \text{Bern}(p)$.
10:         **if** $C_t = 1$ **then**
11:             Set $\widehat{Y}_t = \bot$, observe $Y_t$.
12:             $\mathcal{U}_t \leftarrow \{f : f(X_t) \in \{\bot, Y_t\}\}$
13:             $\mathcal{V}_{t+1} = \mathcal{V}_t \cap \mathcal{U}_t$.
14:         **else**
15:             Pick $\widehat{Y}_t \in \widehat{\mathcal{Y}}_t \setminus \{\bot\}$.
16:             $\mathcal{V}_{t+1} \leftarrow \mathcal{V}_t$.

---

Next, since we are incentivised to minimise the total number of abstentions, it behooves us to play non-abstaining actions whenever possible. However, this puts us in a bind, since feedback is produced only when we play an abstaining action. Taking inspiration from [CLS05], we abstain at a rate $p$ by tossing a biased 'exploratory coin', $C_t$, abstaining when $C_t = 1$, and otherwise playing any non-abstaining action in $\widehat{\mathcal{Y}}_t$. Clearly, such a strategy can incur at most $pT$ excess abstention regret in expectation. Mistakes made by this strategy are controlled via the following 'adversarial law of large numbers' (ALLN).

**Lemma 1.** *Let $\{\mathscr{F}_t\}_{t=1}^\infty$ be any filtration, and $\{U_t\}_{t=1}^\infty, \{B_t\}_{t=1}^\infty$ be $\{\mathscr{F}_t\}$-adapted binary processes, such that $B_t \sim \text{Bern}(p)$, $p < 1/2$ is jointly independent of $\mathscr{F}_{t-1}, U_t$ for each $t$. Let $W_t = \sum_{s \leq t} U_s$,*

*and $\widetilde{W}_t = \sum_{s\le t} U_s B_s$. For any $\delta \in (0, 1/\sqrt{e})$,*

$$\mathbb{P}\left(\exists t : \widetilde{W}_t \le 1, W_t > \frac{8\log(1/\delta)}{p}\right) \le \delta.$$

The above is argued in §A using a self-normalised martingale tail inequality [HRMS20]. We note that this self-normalisation is critical, and without this techniques such as Freedman's inequality yield an extraneous $\sqrt{T}$ factor in the bounds that is untenable for our purposes. The same argument, along with the shaping technique of Howard et al. [HRMS18] yields a Bernstein-type law of iterated logarithms that controls $|W_t - \widetilde{W}_t/p|$ at a level $\tilde{O}(1/p + \sqrt{W_t/p \log\log t})$, which should be useful more broadly. This full version (presented in §A) further shows that the 'Bernoulli-sampler' [BY20; Alo+21] offers a continuous approximation in the sense of Ben-Eliezer & Yogev [BY20], but with the error for sets of low incidence flattened as expected due to Bernstein's inequality.

For our purposes, the point of Lemma 1 is to allow us to argue that no matter what the adversary does, if we uniformly abstain at a rate $p$, then we will 'catch' any mistake-prone function before it makes $O(1/p)$ mistakes. Exploiting a union bound, this in turn means that with high probability, any such function will fall out of the version space $\mathcal{V}_t$ before it has incurred much more than $\log N/p$ mistakes. Since the label produced by Algorithm 1 must equal $f(X_t)$ for *some* $f$ in the version space, we can infer that the number of mistakes the learner makes is at most the number of times any function in the version space is wrong. Using the Lemma yields a bound of $\widetilde{O}(1/p)$ on the number of mistakes that any functions in the version space can have ever made, and since there are only $N$ possible functions, in total the number of mistakes the learner can make is bounded as $\widetilde{O}(N/p)$. More formally, the argument, presented in §B, argues this for a single function $f \in \mathcal{F}$ by instantiating the lemma with $\mathscr{F}_t = \sigma(\mathscr{F}_t = \sigma(\mathscr{H}_t^{\mathfrak{A}}), B_t = C_t$, and $U_t^f := \mathbb{1}\{f(X_t) \notin \{\perp, Y_t\}\}$. The resulting $\widetilde{W}_t^f$ is the number of mistakes $f$ is *observed* to have made, and $f \in \mathcal{V}_t$ if and only if $\widetilde{W}_t^f = 0$[2]. Along with a use of Bernstein's inequality to control $A_T$ this yields the result below.

**Theorem 2.** *Algorithm 1 instantiated with $p < 1/2$, and run against an adaptive adversary, attains the following with probability at least $1 - \delta$ over the randomness of the learner and the adversary:*

$$M_T \le \frac{9N\log(2N/\delta)}{p}$$
$$A_T - A_T^* \le pT + \sqrt{2p(1-p)T\log(2/\delta)} + 2\log(2/\delta).$$

*In particular, taking $p = \sqrt{N/T}$ yields the symmetric regret bound*
$$\max(M_T, A_T - A_T^*) \lesssim \sqrt{NT}\log(N/\delta).$$

We conclude with a few remarks.

**Achievable rates**: Taking $\delta = 1/T$, and varying $p$ in $(\log T/T, 1]$ gives the rates attainable by VUE

**Corollary 3.** *All rates $(\alpha, \mu)$ such that $\alpha > 0, \alpha + \mu > 1$ are achievable against adaptive adversaries.*

These rates are tight - as expressed in Corollary 6, rates such that $\alpha + \mu < 1$ are not achievable even against stochastic adversaries. The Pareto frontier is therefore the line $\alpha + \mu = 1$.

**Dependence on $N$**: It should be noted that the dependence on the number of functions, $N$, in Thm. 2 is polynomial, as opposed to the more typical logarithmic dependence on the same in online classification. The problem of characterising this dependence appears to be subtle, and we do not resolve the same. In the following section, we explore schemes that improve this aspect, but at a cost - §4 yields logarithmic dependence against stochastic adversaries, while §5 gives a scheme that has a logarithmic dependence against adaptive adversaries, but worse dependence with $T$.

It is worth stating that the analysis above is tight for Algorithm 1 - consider the domain $\mathcal{X} = [1 : N]$, and the class $\mathcal{F} = \{f_t : t \in [0 : N]\}$ such that $f_t(x) = \perp$ if $x \le t$ and $= 1$ if $x > t$. Now consider an adversary that chooses a $t^*$ in advance, and presents the contexts $1$ $T/N$ times, $2$ $T/N$ times and so on, labelling contexts smaller than $t^*$ as $0$, and contexts larger than $t^*$ as $1$. Notice that in each case, there is exactly one function in $\mathcal{V}_t$ that does not abstain. The scheme above incurs $\Omega(pT(1 - t^*/N))$ excess abstention, and $\Omega(t^*/p)$ mistakes, and linearly large $t^*$ form a tight example. Of course, this is not a lower bound on this problem, and the question of the optimal dependence on $N$ remains open.

---

[2]This argument only needs control for the case $\widetilde{W}_t = 0$. The $\le 1$ in Lemma 1 is exploited in §5.2.

**Hedge-Type Schemes** The natural approach of proceeding by weighing the cost of abstention versus a mistake, and running a hedge-type scheme on an importance-estimate of the resulting loss does not lead to tight rates - the scheme MIXED-LOSS-PROD of §5 pursues precisely this strategy, and the worse case symmetric regret bounds that standard analyses lead to scale as $T^{2/3}$ instead of as $T^{1/2}$ as for VUE (Cor. 8). This may be due to the fact certain-error prone classifiers in $\mathcal{F}$ may have very low abstention rates, and thus overall large weight, and it is unclear how to eliminate this behaviour.

## 4 The Stochastic Case

This section argues that the regret bounds of Thm. 2 can be improved to behave logarithmically in $N$ in the stochastic setting. There are a couple of issues with Algorithm 1 that impede a better analysis in the stochastic case. The first, and obvious, one is that how $\widehat{Y}_t$ is chosen is not specified. More subtly, the fact that the scheme insists on playing non-abstaining actions whenever possible makes it difficult to control the number of mistakes without a polynomial dependence on $N$.

We sidestep these issues in Algorithm 2 by maintaining a law $\pi_t$ on functions in $\mathcal{V}_t$ that only depends on $\mathscr{H}_{t-1}^{\mathfrak{L}}$, and predicting by setting $\widehat{Y}_t = f(X_t)$ for $f_t \sim \pi_t$. Notice that playing this way it is possible that we abstain on $X_t$ even if the exploratory coin comes up tails. We control mistakes by arguing that very error-prone functions are all quickly eliminated (due to the stochasticity), and using the property that $\pi_t$ does not depend on $X_t$ to limit the mistakes incurred up to such a time. Abstention control follows by choosing $\pi$ according to a strategy that favours $f$s with small overall abstention rate over the history. In Algorithm 2, we use a version of the PROD scheme of [CMS07] to set weights, analysed with shrinking decision sets. The following is shown along these lines in §C.

**Theorem 4.** *Algorithm 2, run against stochastic adversaries with $\eta = p$, attains the regret bounds*

$$\mathbb{E}[M_T] \leq 8 \frac{\log T \log(NT)}{p}, \quad and \quad \mathbb{E}[A_T - A_T^*] \leq pT + \frac{\log N}{p}.$$

We note that VUE-PROD also enjoys favourable bounds in the adversarial case - mistakes are bounded as $\tilde{O}(N/p)$, and abstention regret as in the above result. This is in contrast to simpler follow-the-versioned-leader type schemes that also satisfy similar bounds as Thm. 4 in the stochastic case. Also note that the above cannot attain rates such that $\alpha \leq 1/2$, an inefficiency introduced due to the conditional independence of $\pi_t$ and $X_t$.

Finally, we show a lower bound. The statement equates stochastic adversaries with their laws.

**Theorem 5.** *If $\mathcal{F}$ contains two functions $f_1, f_2$ such that there exists a point $x$ for which $f_1(x) = \perp \neq f_2(x)$, then for every $\gamma \in [0, 1/2]$, there exists a pair of laws $P_1^\gamma, P_2^\gamma$ such that any learner that attains $\mathbb{E}_{P_1^\gamma}[A_T - A_T^*] = K$ must incur $\mathbb{E}_{P_2^\gamma}[M_T] \geq \gamma(e^{-2\gamma K}T - K)$.*

---

**Algorithm 2** VUE-PROD

1: **Inputs**: $\mathcal{F}, p$, Learning rate $\eta$.
2: **Initialise**: $\mathcal{V}_1 \leftarrow \mathcal{F}, \forall f, w_1^f \leftarrow 1$.
3: **for** $t \in [1 : T]$ **do**
4:     Sample $f_t \sim \pi_t = \frac{w_t^f \mathbb{1}\{f \in \mathcal{V}_t\}}{\sum_{f \in \mathcal{V}_t} w_t^f}$.
5:     Toss $C_t \sim \text{Bern}(p)$.
6:     $\widehat{Y}_t \leftarrow \begin{cases} \perp & C_t = 1 \\ f_t(X_t) & C_t = 0 \end{cases}$.
7:     $\mathcal{V}_{t+1} \leftarrow \mathcal{V}_t$.
8:     **if** $C_t = 1$ **then**
9:         $\mathcal{U}_t \leftarrow \{f : f(X_t) \in \{\perp, Y_t\}\}$
10:         $\mathcal{V}_{t+1} = \mathcal{V}_t \cap \mathcal{U}_t$.
11:     **for** $f \in \mathcal{V}_{t+1}$ **do**
12:         $a_t^f \leftarrow \mathbb{1}\{f(X_t) = \perp\}$
13:         $w_{t+1}^f \leftarrow w_t^f \cdot (1 - \eta a_t^f)$.

---

Thus, if a $(\varphi, \psi)$ regret bound with $\sup \frac{\varphi}{T} < \frac{1}{2e^2}$ is achievable, then $\varphi \cdot \psi = \Omega(T)$. Indeed, using the above with $\gamma = 1/\varphi(T, N)$, gives $\mathbb{E}_{P_1}[A_T - A_T^*] = K \leq \varphi(T, N)$, and so $\psi(T, N) \geq \mathbb{E}_{P_2}[M_T] \geq \frac{T}{\varphi(T,N)} e^{-2K/\varphi(T,N)} - 1$. This proves the following.

**Corollary 6.** *If $(\alpha, \mu) \in [0, 1]^2$ is such that $\alpha + \mu < 1$, then an $(\alpha, \mu)$ regret rate is not achievable against stochastic adversaries, and, a fortiori, against adaptive adversaries.*

## 5 Reducing the dependence of regret bounds on $N$ in the adversarial case

This section concentrates on improving the $N$-dependence of regret bounds in the adversarial case via two avenues. The first improves this dependence to $\log(N)$ by running PROD with a weighted loss, but at the cost of increasing $T$ dependence. This holds greatest relevance when $T$ is bounded as a polynomial of $N$, which is of interest because $N$ can be quite large even in reasonable settings - e.g., a discretisation of $d$-dimensional hyperplanes induces $N = \exp(Cd)$. The second approach

considers the case when the set of possible contexts, i.e. $\mathcal{X}$ is not too large. While in this case, $N$ can be as large as $(|\mathcal{Y}| + 1)^{|\mathcal{X}|}$, we show bounds depending only linearly on $|\mathcal{X}|$.

## 5.1 Weighted PROD

We continue the uniform exploration, but play according to the PROD method, with the loss

$$\ell_t^f := C_t \mathbb{1}\{f(X_t) \notin \{\bot, Y_t\}\} + \lambda \mathbb{1}\{f(X_t) = \bot\},$$

where $\lambda$ both trades-off the relative costs of mistakes and abstentions, in the vein of the fixed cost Chow loss, and accounts for the sub-sampling of the mistake loss.

The analysis of this scheme, presented in §D exploits the quadratic bound of PROD due to [CMS07] to control the sum $\mathbb{E}[pM_T + \lambda(A_T - pT)]$ by $\min_g \log N/\eta + \sum \eta(\ell_t^g)^2$, where the expectation is only over the coins $C_t$, and the $-pT$ term is due to the extra abstentions due to the exploratory coin. The key observation is that

---

**Algorithm 3** MIXED-LOSS-PROD

1: **Inputs**: $\mathcal{F}$, Exploration rate $p$, Learning rate $\eta$.
2: **Initialise**: $\forall f \in \mathcal{F}, w_1^f \leftarrow 1$.
3: **for** $t \in [1 : T]$ **do**
4:      Sample $f_t \sim \pi_t = w_t^f / \sum w_t^f$.
5:      Toss $C_t \sim \mathrm{Bern}(p)$.
6:      **if** $C_t = 1$ **then**
7:          $\widehat{Y}_t \leftarrow \bot$
8:      **else**
9:          $\widehat{Y}_t \leftarrow f_t(X_t)$
10:      $\forall f \in \mathcal{F}$, evaluate $\ell_t^f$
11:      $w_{t+1}^f \leftarrow w_t^f(1 - \eta \ell_t^f)$.

---

since $f^*$ makes no mistakes, $\sum (\ell_t^{f^*})^2 = \lambda^2 A_T^*$, and so taking $g = f^*$, and exploiting the weight allows us to separately control the regrets in terms of $A_T^*$.

**Theorem 7.** *Algorithm 3, when run against adaptive adversaries with $\eta = 1/2, \lambda \leq p$, attains*

$$\mathbb{E}[M_T] \leq \frac{2 \log N}{p} + \frac{2\lambda}{p} \mathbb{E}[A_T^*], \quad \text{and} \quad \mathbb{E}[A_T - A_T^*] \leq pT + \frac{2 \log N}{\lambda}.$$

### 5.1.1 Rates

Theorems 4 and 7 show regret bounds with logarithmic dependence in $N$. The following concept separates rates attainable with this advantageous property from those with worse $N$-dependence.

**Definition** (Logarithmically Achievable Rates) *We say that rates $(\alpha, \mu)$ are logarithmically achievable against adversaries from a class $\mathscr{C}$ if there exists a learner that attains a $(\psi, \varphi)$-regret against such adversaries for $\psi, \varphi$ that witness the rate $(\alpha, \mu)$, and satisfy that for every fixed $T$, $\max(\varphi(T, N), \psi(T, N)) = O(\mathrm{polylog}(N))$ as $N \to \infty$.*

Since $A_T^* \leq T$, choosing $p = T^{-u}, \lambda = T^{-(u+v)}$ in MIXED-LOSS-PROD for any $(u, v) \in [0, 1]^2, u + v \leq 1$ allows us to attain rates of the form $(\alpha, \mu) = (\max(1 - u, u + v), 1 - v)$. Notice that for any fixed $v$, the smallest $\alpha$ so attainable is $1+v/2$. This shows

**Corollary 8.** *Any rate $(\alpha, \mu)$ such that $\alpha + \mu/2 > 1$ is logarithmically achievable against adaptive adversaries.*

The following figure illustrates the worst case achievable rate regions in the three cases considered.

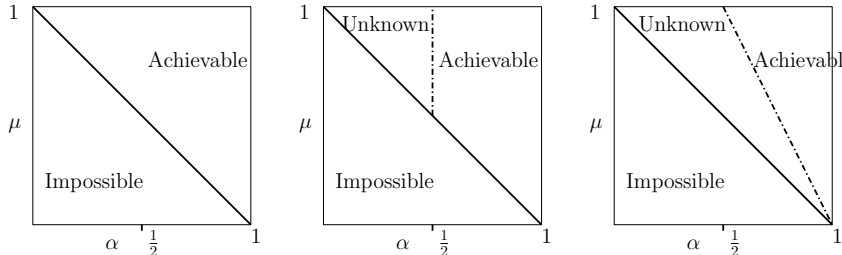

Figure 1: Left shows rates achievable against adaptive adversaries. Middle and right show logarithmically achievable rates against stochastic and adaptive adversaries respectively.

**Adaptive Rates** Observe that if $A_T^* \asymp T^{\alpha^*}$ for some $\alpha^* < 1$, then nominally, the achievable rates can be improved. Indeed, with the parametrisation $p = T^{-u}, \lambda = T^{-u+v}$, we may attain rates of the form $(\alpha, \mu) = (\max(1 - u, u + v), \max(u, \alpha^* - v))$. Further, a given mistake rate $\mu$ can be attained by setting $u = \mu$, and $\alpha^* - v \leq \mu$. With these constraints, the smallest abstention rate attainable is

$$\widetilde{\alpha}(\mu; \alpha_*) = \max\left(1 - \mu, (1 + (\alpha^* - \mu)_+)/2\right),$$

achieved by setting $v = (\alpha_* - \mu)_+, u = \min(1 - (\alpha^* - \mu)_+, 2\mu)/2$. Such rates can in fact be attained adaptively, without prior knowledge of $\alpha^*$. The main bottleneck here is that the quantity $A_T^*$ is not observable. However, every function $g$ that is never *observed* to make a mistake satisfies $\sum (\ell_t^g)^2 = \lambda^2 \sum \mathbb{1}\{g(X_t) = \bot\}$, and such functions are identifiable given $\mathscr{H}_t^{\mathfrak{L}}$. Let

$$B_t^* := \min \sum_{s \leq t} \mathbb{1}\{g(X_s) = \bot\} \quad \text{s.t.} \quad \sum_{s \leq t} C_s \mathbb{1}\{g(X_s) \notin \{\bot, Y_t\}\} = 0.$$

Note that $B_t^*$ grows monotonically, and is always smaller than $A_t^* = \sum_{s \leq t} \mathbb{1}\{f^*(X_t) = \bot\}$. We show the following in §D.1 via a scheme that adaptively sets $p, \lambda$ according to $B_t^*$.

**Theorem 9.** *For any $\alpha^*, \mu, \varepsilon \in (0, 1]$, Algorithm 4 attains, without prior knowledge of $\alpha^*$, any rate of the form $(\widetilde{\alpha}(\mu, \alpha^*) + \varepsilon, \mu + \varepsilon)$ against adaptive adversaries that induce $A_T^* \leq T^{\alpha^*}$ almost surely.*

The rates $\widetilde{\alpha}$ essentially interpolate between the second and third panels of Fig. 1. Concretely the region achieved consists of the intersection of the regions $\{\alpha > 1/2\}, \{\alpha + \mu > 1\}$ and $\{2\alpha + \mu > 1 + \alpha^*\}$, with the last set being active only when $\alpha^* \geq 1/2$.

### 5.2 A $|\mathcal{X}|$-dependent analysis of VUE

We give an alternate mistake analyse for VUE over finite domains. The analysis is slightly stronger: let $\mathbf{y} \in ([1 : K] \cup \{\bot\})^{|\mathcal{F}|}$ be indexed by elements of $\mathcal{F}$, with the '$f$th' entry $\mathbf{y}_f$ reprsents a value that $f$ might take. Consider the resulting partition of $\{\mathcal{X}_\mathbf{y}\}_{\mathbf{y} \in ([1:K] \cup \{\bot\})^\mathcal{F}}$, where each part $\mathcal{X}_\mathbf{y} \subset \mathcal{X}$ contains points that have the same pattern of function values, that is $\mathcal{X}_\mathbf{y} = \{x : \forall f \in \mathcal{F}, f(x) = \mathbf{y}_f\}$. The following argument can be run unchanged by replacing single $x$s in the following by all $x$s in one $\mathcal{X}_\mathbf{y}$. That is, we may replace $|\mathcal{X}|$ in the following Theorem 10 with $|\{\mathcal{X}_\mathbf{y}\}|$. For simplicity, we present the argument for $|\mathcal{X}|$ only.

Denote $\widehat{\mathcal{Y}}_t^x := \{f(x) : f \in \mathcal{V}_t\}$. Notice that after the first time $t$ such that $X_t = x, \widehat{Y}_t = \bot$, we will remove from the version space all classifiers that did not abstain or output the correct classification at time $t$. Thus if we define $y^x \in [1 : K]$ to be $Y_t$, then for all subsequent times, $\widehat{\mathcal{Y}}_t^x \subset \{\bot, y^x\}$. As a result, if we observe two mistakes at any given $x$, then we cannot make any more mistakes at a subsequent time $t'$ with $X_{t'} = x$, because the only remaining decision in $\widehat{\mathcal{Y}}_{t'}^x$ must be $\bot$.

We may now proceed in much the same way as §3 - instantiate $U_t^x = \mathbb{1}\{X_t = x, \widehat{Y}_t \notin \{\bot, Y_t\}\}$, $B_t = C_t$, and union bound over the $x$s. Then $|\widehat{\mathcal{Y}}_t^x| \geq 2$ if and only if $\widetilde{W}_t^x \leq 1$, and, invoking Lemma 1, up to such a time at most $W_t^x = O(\log |\mathcal{X}|/p)$ mistakes may be made on instances such that $X_t = x$. But then totting up, we make at most $O(|\mathcal{X}| \log |\mathcal{X}|/p)$ mistakes, as encapsulated below

**Theorem 10.** *Algorithm 1 instantiated with $p \leq 1/2$ and run against an adaptive adversary, attains the following with probability at least $1 - \delta$ over the randomness of the learner and the adversary:*

$$M_T \leq \frac{9|\mathcal{X}| \log(2|\mathcal{X}|/\delta)}{p}$$
$$A_T - A_T^* \leq pT + \sqrt{2p(1-p)T \log(2/\delta)} + 2\log(2/\delta).$$

Along with the bound itself, the above result makes a couple of points regarding the characterisation of $N$-dependence of the regrets in online selective classification. Firstly, it suggests that efficient analyses, and possibly schemes, must incorporate the structure of $\mathcal{X}$; and secondly it shows that constructions that attempt to show superlogarithmic in $N$ lower bounds must have both $N$ and $|\mathcal{X}|$ large, and thus typical strategies placing a very rich class on a small domain will not be effective.

## 6 Experiments

We evaluate the performance of Algorithm 2 on two tasks - CIFAR 10 [KH09], and GAS [Ver+12] - see §E for details of implementation, and here for the relevant code. The former represents a setting where an expert can be adaptively invoked, which we treat by providing the true labels of the classes upon abstention. The second case is more explicitly an adaptive feature selection task - the GAS dataset has features from 16 sensors, and we train one model, $g$, on all of this data, while the selective classification task operates on data from the first 8 sensors only, and receives the output of $g$ when abstaining. The standard accuracies of the model classes we implement are $\sim 90\%$ on CIFAR-10,

and $\sim 77\%$ on GAS. In both cases, a training set is used to learn a parameterized family of selective classifiers, $f_{\mu,t}$. The hyperparameters $(\mu, t)$ provide control over various levels of accuracy and abstention. For training, we leverage a recent method [GKS21] that yields such a parameterisation, which is discretised to get $N = 600$ of these functions to form our class $\mathcal{F}$. We then sequentially classify the test datasets of each of the tasks.

One subtlety with the setting is that none of the selective classifiers in $\mathcal{F}$ actually make no mistakes. To avoid the trivialities emerging from this, we relax the versioning condition to only drop classifiers that are seen to make mistakes on at least $\varepsilon N_t + \sqrt{2\varepsilon N_t}$ mistakes at time $t$, where $N_t$ is the number of times feedback was received up to time $t$, and the second term handles noise. Additionally, if it turns out that all functions in $\mathcal{V}_t$ are wrong on a particular observed instance, we ignore this feedback (since such an error is unavoidable). Such variations of 'relaxed versioning' are natural ideas when extending the present problem to the one where the competitor may be allowed to make non-zero mistakes, although its analysis is beyond the scope of this paper. The scheme's viablility in this extended setting with only simple modifications indicates the practicality of such strategies.

Below, we take the competitor to be the function that makes the fewest mistakes, denoted as $M_T^*$. If there is more than one such function, we take the one that makes the fewest abstention to get $A_T^*$. We measure *excess mistakes* $M_T - M_T^*$ and excess abstentions $A_T - A_T^*$ with respect to this competitor.

**Behaviour of regrets with the length of the game** Fig. 2 presents the excess mistakes as a fraction of $T$ for the two datasets, i.e. $M_T - M_T^*/T$, as $T$, is varied. The learners are all instantiated with the exploration rate $p = 1/\sqrt{T}$. We observe that the excess abstentions are negative (or near-zero) over this range (see Fig. 4 in §E). Therefore we do not plot these below (the orange line is MMEA, see below). We note that the relative mistakes stay below $\sqrt{2 \log N / T}$, bearing out the theory.

**Achievable Operating Points of Mistakes and Abstentions** Fig. 3 shows the mistake and abstention rates attainable by varying $p$ and $\varepsilon$, while holding $T$ fixed at 500 (which is large enough to show long-run structure, but small enough allow fast experimentation). Concretely, we vary these linearly for 20 values of $p \in [0.015, 0.285]$, and 10 values of $\varepsilon \in [0.001, 0.046]$. The resulting values represent operating points that can be attained by a choice of $p, \varepsilon$. The same plot includes lines that represent the operating points when the scheme is run with $\varepsilon = 0.001$, the smallest value we take. Note that in practice, the best choices of $\varepsilon, p$ may be data dependent, and choosing them in an online way is an interesting open problem (also see §E.6).

**The Price of Being Online** We characterise this in two ways beyond the excess mistakes.

• In Fig. 2, we also plot the 'mistake-matched excess abstention' (MMEA). This is defined as follows - if the scheme concludes with having made $M_T$ mistakes, we find, in hindsight, the classifier that minimises the number of abstentions, subject to making at most $M_T$ mistakes. The MMEA is the excess abstention of the learner over those of this relaxed competitor, and represents how many fewer abstentions a batch learner would make if allowed to make as many mistakes as the online learner. Notice that this MMEA remains well controlled in Fig. 2, and appears to scale as $O(\sqrt{T})$.
• In Fig. 3, we also plot the post-hoc operating points of the classifiers in $\mathcal{F}$ as black triangles. This amounts to plotting the optimal abstentions amongst classifiers that make at most $m$ mistakes, varying $m$.[3] We note that the red operating points of the scheme get close to the black frontier, illustrating that the inefficiency due to being online is limited. As the time-behaviour of MMEA in Fig.2 illustrates, the inefficiency is expected to grow sublinearly with $T$, and to thus vanish under amortisation.

## 7    Discussion

Online selective classification offers a primitive that has relevance to both safety-critical and resource-limited settings. In the paper, we highlighted the role of long-term abstentions in such situations, and studied this problem under the feedback limitation that labels are only provided when the system abstains, which is the only time high-complexity evaluation would be invoked in a selective classification system. When working with a finite class of model, we identified a simple scheme that provides a tight (in terms of $T$) trade-off between mistakes and excess abstentions against adaptive adversaries. We further discussed two schemes that improve upon the dependence of the same on the size of the model class - tightly against stochastic adversaries, and at the cost of some rate

---

[3]Observe that the MMEA corresponds to the horizontal distance between a red-point with $m$ mistakes, and the left-most black point with $y$-coordinate under $m$.

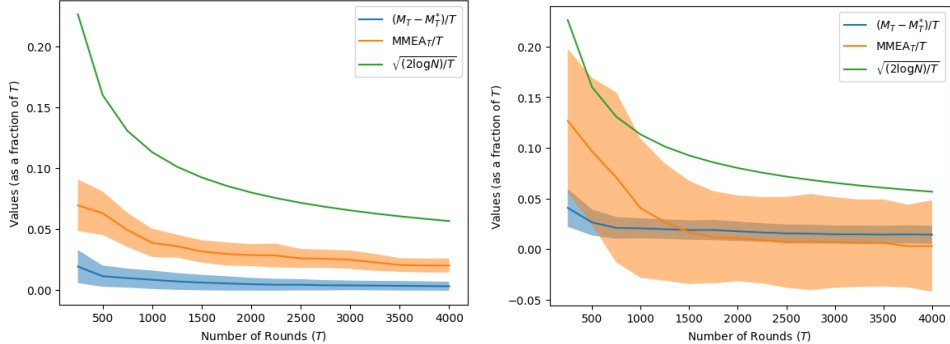

Figure 2: $M_T - M_T^*$, and MMEA as fractions of $T$, as the number of rounds $T$ is varied for CIFAR-10 (left) and GAS (right). The plots are averaged over 100 runs, and one-standard-deviation error regions are drawn.

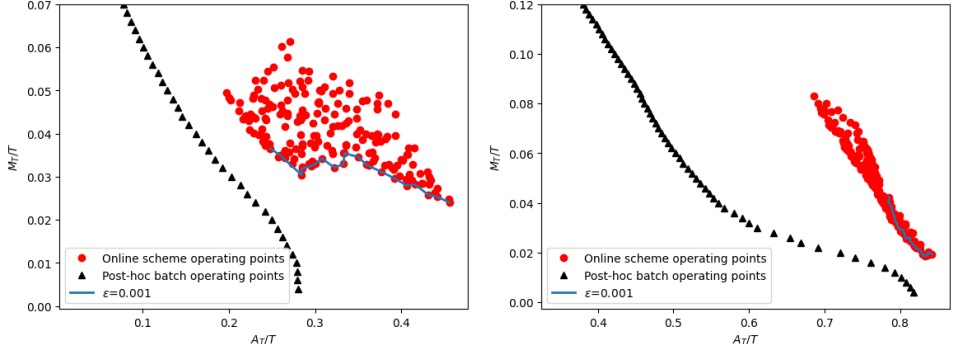

Figure 3: Operating points for our scheme as $\varepsilon$ and $p$ are varied are represented as red dots (for CIFAR-10 in the left, and GAS in the right). The black triangles represent operating points obtained by batch learning with the benefit of full feedback. The blue lines interpolate points obtained by varying $p$ for $\varepsilon = 0.001$ Points are averaged over 200 runs. Note that the values are raw mistakes and abstentions, and not regrets.

performance against adaptive adversaries. Together, these schemes and analyses provide some basic foundations for the problem when competing against no-mistake models. Additionally, we carried out empirical studies that validate the scheme in the stochastic case, and demonstrate that with minor modifications, the scheme is resilient to the situation where no selective classifier in the model class is mistake-free. A number of interesting questions remain open, and we discuss a few of these below.

Perhaps the most basic question left open by the above study is how the minimax regrets against adaptive adversaries depend on $N$. Along with being a basic scientific question, this issue has implications for whether the results can be extended to infinite classes. Indeed, under assumptions of bounded combinatorial dimensions, the VUE-PROD and MIXED-LOSS-PROD schemes can be extended to infinite model classes, but the basic technique to do so yields trivial bounds for VUE due to the linear dependence on $N$. If this dependence could be improved to logarithmic, the extension to model classes with finite (multiclass versions of) Littlestone dimension would be immediate.

A practically relevant and theoretically interesting direction is online SC but where the competitor can make non-zero mistakes. This can be set up in at least two ways - either an error parameter $\varepsilon$ is given to the learner, which must ensure that both notions of regret are small against competitors that make at most $\varepsilon T$ mistakes; or, no explicit error parameter is specified, and the learner is required to compete against the least mistake-prone model in a given set (similarly to §6). Both settings raise new challenges, since one must relax the notion of versioning used in the above work for related schema to be viable. The latter setting raises a further issue of how one can adapt to the mistake rate of the competitor. Also of practical relevance is the case where abstentions are not equally penalised, but have some variable cost. Here too, one can study variants of signalling regarding whether the cost of abstention is available before or only after an abstaining decision is made.

Finally, we observe that while tight, the random exploration technique is somewhat unsatisfying, and practically a context-adapted abstention strategy is likely to offer meaningful advantages over it. In analogy with the exploration in label-efficient prediction, one direction towards exploring context-aware methods is to study more concrete structured situations, such as linear models with noisy feedback that are popular in the investigation of online selective sampling.

**Acknowledgements** Our thanks to Tianrui Chen for helpful discussions.

**Funding Disclosure** This research was supported by the Army Research Office Grant W911NF2110246, the National Science Foundation grants CCF-2007350 and CCF-1955981, ARM Research Inc, and the Hariri Data Science Faculty Fellowship Grants. Additional revenues related to this work: AC was a visiting researcher at Google when this work was completed.

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
