# A  An Adversarial Anytime Uniform Law of Large Numbers For Probing Binary Sequences

## A.1  Proofs of Lemma 1

We begin with a simple lemma that underlies the remaining argument. Below, $\kappa$ is chosen so that $\kappa''(0) = 1$.

**Lemma 11.** *Let $\mathscr{F}_t, U_t, B_t, W_t, \widetilde{W}_t$ be as in Lemma 1. Let $\overline{p} = 1 - p$. Then for any $\eta \in \mathbb{R}$, the process*

$$\xi_t^\eta := \exp\left( \eta(W_t - \widetilde{W}_t/p) - \kappa(\eta)V_t \right)$$

*is a non-negative, $\mathscr{F}_t$-adapted martingale, where*

$$V_t = \frac{\overline{p}}{p} W_t,$$

$$\kappa(\eta) = \frac{p}{\overline{p}} \log\left( p e^{-\eta \overline{p}/p} + \overline{p} e^\eta \right).$$

*Proof.* The nonnegativity of $\xi_t^\eta$ is trivial, and it is $\mathscr{F}_t$-adapted since it is a deterministic function of the adapted processes $W_t, \widetilde{W}_t$. We need to argue that $\xi$ is a martingale. To this end, observe that since $W_t = \sum_{s<t} U_s, \widetilde{W}_t = \sum_{s<t} U_s B_s$,

$$\xi_t^\eta = \xi_{t-1}^\eta \cdot \exp\left( \eta U_t(1 - B_t/p - \overline{p}\kappa(\eta)/p) \right).$$

Due to the independence of $B_t$ from $\sigma(U_t, \mathscr{F}_{t-1})$, we have

$$\mathbb{E}[\exp\left( \eta U_t(1 - B_t/p) \right) | \mathscr{F}_{t-1}, U_t]$$

$$= \left( p e^{-\eta U_t \overline{p}/p} + \overline{p} e^{\eta U_t} \right)$$

$$\overset{*}{=} \left( p e^{-\eta \overline{p}/p)} + \overline{p} e^\eta \right)^{U_t} = \exp\left( \frac{\overline{p}}{p} U_t \kappa(\eta) \right),$$

where the equality marked $*$ exploits the fact that $U_t$ is $\{0, 1\}$-valued. Rearranging, we have

$$\mathbb{E}\left[ \exp\left( \eta U_t(1 - B_t/p) - \frac{\overline{p}}{p} U_t \kappa(\eta) \right) \middle| \mathscr{F}_{t-1}, U_t \right] = 1,$$

and exploiting the tower rule, we conclude that

$$\mathbb{E}[\xi_t^\eta | \mathscr{F}_{t-1}] = \xi_{t-1}^\eta \mathbb{E}\left[ \mathbb{E}\left[ \exp\left( \eta U_t(1 - B_t/p) - \frac{\overline{p}}{p} U_t \kappa(\eta) \right) \middle| \mathscr{F}_{t-1}, U_t \right] \middle| \mathscr{F}_{t-1} \right] = \xi_{t-1}^\eta. \quad \square$$

The following argument heavily exploits the techniques of Howard et al. [HRMS20], and assumes familiarity with the same. It also exploits the property that only the upper tail of $\Delta_t$ is being controlled, although this is extended in the following section.

*Proof of Lemma 1.* We define the deviation of $W_t$ from $\widetilde{W}_t$ as

$$\Delta_t := W_t - \frac{\widetilde{W}_t}{p}.$$

Notice that $\Delta_0 = 1$. As a result of the above lemma, $\Delta_t$ is a 1-sub-$\kappa$ process with the associated variance process $V_t$, in the sense of Definition 1 of Howard et al. [HRMS20]. In particular, since $\kappa$ is the (normalised) cumulant generating function of a centred Bernoulli random variable taking values $\{-\overline{p}/p, 1\}$, the process is sub-binary. Further, since $p < 1/2, \overline{p}/p > 1$, and thus the process is sub-gamma, with the scale parameter $c = 0$. [HRMS20, §3.1, and Prop. 2].

We can thus invoke the line-crossing inequality of Corollary 1, part c) of Howard et al., instantiated with $c = 0$ to find that for any $x, m > 0$

$$\mathbb{P}\left( \exists t : \Delta_t \geq x + \mathfrak{s}(x/m)(V_t - m) \right) \leq \exp\left( -\frac{x^2}{2m} \right),$$

where [HRMS20, Table 2]

$$\mathfrak{s}(x/m) = \frac{x}{2m}.$$

Plugging these in, we observe that

$$\mathbb{P}\left(\exists t : \Delta_t \geq \frac{x}{2} + \frac{x}{2m}V_t\right) \leq \exp\left(-\frac{x^2}{2m}\right).$$

Now notice that if $V_t \geq m$, then $x/2 + (x/2m)V_t \leq (x/m)V_t$. Therefore, we can conclude that

$$\mathbb{P}\left(\exists t : \Delta_t \geq \frac{x}{m}V_t, V_t \geq m\right) \leq \exp\left(-\frac{x^2}{2m}\right),$$

and substituting $V_t = \frac{\overline{p}}{p}W_t, \Delta_t = W_t - \widetilde{W}_t/p$,

$$\mathbb{P}\left(\exists t : \frac{\widetilde{W}_t}{p} \leq \frac{mp - x\overline{p}}{pm}W_t, W_t \geq \frac{pm}{\overline{p}}\right) \leq \exp\left(-\frac{x^2}{2m}\right).$$

Now, if we choose $m = \frac{\overline{p}}{p}(x + 1/p)$ it follows that

$$\forall W_t \geq \frac{p}{\overline{p}}m, \frac{pm - x\overline{p}}{mp}W_t \geq \frac{1}{p},$$

and thus

$$\mathbb{P}\left(\exists t : \frac{\widetilde{W}_t}{p} \leq \frac{1}{p}, W_t \geq \frac{1}{p} + x\right) \leq \exp\left(-\frac{px^2}{2(1/p + x)\overline{p}}\right),$$

and choosing $x \geq 1/p$ further ensures that

$$\mathbb{P}\left(\exists t : \widetilde{W}_t \leq 1, W_t \geq 2x\right) \leq \exp\left(-\frac{px}{4\overline{p}}\right).$$

Now, setting $x = \max\left(\frac{1}{p}, \frac{4\overline{p}}{p}\log(1/\delta)\right)$ leaves us with

$$\mathbb{P}\left(\exists t : \widetilde{W}_t \leq 1, W_t \geq \max\left(\frac{2}{p}, \frac{8\overline{p}}{p}\log(1/\delta)\right)\right) \leq \delta.$$

The conclusion follows on observing since $p < 1/2, 8\overline{p} \geq 4$, and thus, for $\log(1/\delta) \geq 1/2, \frac{2}{p} \leq 8\frac{\overline{p}}{p}\log(1/\delta)$. $\qquad\square$

## A.2 An improved ALLN via a Self-Normalised Law of Iterated Logarithms

The line-crossing inequalities we utilised in the previous subsection can be stitched together, by picking an exponentially increasing set of $x$s, and optimising the $m$s at each, to yield a curve crossing inequality, which in effect determines a curve that the deviations are unlikely to cross. We use the results of Howard et al. [HRMS18] that produce non-asymptotic constructions.

For our purposes, note that the processes $\Delta_t$ and $-\Delta_t$ are both sub-Gamma with variance process $V_t$, with the scale parameters $c_+ = 0$ and $c_- = \frac{1}{3} \cdot \frac{1-2p}{p}$ respectively. The former property is useful for controlling the upper deviations of $\Delta_t$, and the latter for the lower deviations. Note that since the scale parameter $c_+$ is 0, the upper tails in the following can be improved, but for ease of presentation we will just set $c = |c_+| = c_-$ in the following.

Using Theorem 1 of Howard et al. [HRMS18] twice - for $\Delta_t$ and $-\Delta_t$, and instantiating it with $\eta = e, h(k) = \frac{\pi^2 k^2}{6}$ yields that for the sub-gamma process $\Delta_t$ with scale parameter $\leq c$, and variance process $V_t$, and any constant $m > 0$, and for the functions

$$S_{m,\delta}(v) = 2\sqrt{v\ell_{m,\delta}(v)} + c\ell_{m,\delta}(v),$$

$$\ell_{m,\delta}(v) = \log\frac{\pi^2}{6} + 2\log\log\frac{v}{m} + \log\frac{2}{\delta},$$

the following bound holds true

$$\mathbb{P}(\exists t : |\Delta_t| \geq \mathcal{S}_{m,\delta}(\max(V_t, m)) \leq \delta.$$

The curve $S(\max(V_t, m))$ can be simplified upon observing that

$$\{\exists t : V_t \geq m, |\Delta_t| \geq \mathcal{S}_{m,\delta}(V_t)\} \subset \{\exists t : |\Delta_t| \geq \mathcal{S}_{m,\delta}(\max(V_t, m))\}.$$

With the above in hand, set $m = \overline{p}/p$, so that $W_t \geq 1 \iff V_t \geq m$, and observe that $\log(\pi^2/6) <$ 1. The following bound is immediate upon recalling that $V_t = \frac{\overline{p}}{p} W_t, \Delta = W_t - \widetilde{W}_t/p$.

**Theorem 12.** *In the setting of Lemma 1,*

$$\mathbb{P}\left(\exists t : W_t \geq 1, |W_t - \widetilde{W}_t/p| \geq 2\sqrt{\frac{\overline{p}W_t}{p}\left(\log\frac{2e}{\delta} + 2\log\log W_t\right)} + \frac{\log{2e/\delta} + 2\log\log W_t}{3p}\right) \leq \delta.$$

Technically, the $\log\log W_t$ is not always defined in the above. This should be read as $\log(\max(1, \log W_t))$ to handle edge cases - alternately, it can be handled by replacing $W_t \geq 1$ by $W_t \geq 3 > e$ in the above.[4]

Notice that the bound above has the correct form when taking into account the behaviour of binomial tails, which $\widetilde{W}_t$ behaves like. Indeed, if $W$ is some natural number valued random variable, and $\widetilde{W}|W \sim \text{Bin}(W, p)$, then Bernstein's inequality [BLM13, Ch. 2] states that

$$\mathbb{P}\left(|W - \widetilde{W}/p| \geq C\sqrt{\overline{p}\frac{W}{p}\log(2/\delta)} + C\log(2/\delta)\right) \leq \delta,$$

which entirely parallels the form of the above theorem, barring the $\log\log W_t$ blowup due to the uniformity over time.

The above analysis was inspired by studying the recent work of Ben-Eliezer and Yogev [BY20], on adversarial sketching - their goal was to maintain an estimate of the incidence of a process within a given set (and more generally, within sets in a given system) while using limited memory, and they analysed a similar sampling approach, showing via an application of Freedman's inequality that [BY20, Lemma 4.1]

$$\mathbb{P}\left(|W_T - \widetilde{W}_T/p| \geq C\sqrt{\frac{T}{p}\log(2/\delta)} + C\frac{\log(2/\delta)}{p}\right) \leq \delta.$$

This essentially amounts to using the crude bound $W_T \leq T$. The same paper, in Theorem 1.4 and associated lemmata argues that the Reservoir Sampler [BY20, §2] of size $\sim pT$ controls deviations uniformly over time at scale $\sqrt{\frac{T}{p}\log\frac{\log T}{\delta}}$, and it was asserted that the Bernoulli Sampler cannot attain such a 'continuous robustness'[BY20, §1]. The above result improves upon this in a few ways - firstly, the result applies to the simpler Bernoulli sampler, and improves the deviation control to $O(\sqrt{W_t})$ instead of $O(\sqrt{T})$. This has the further advantage that if one is concerned with the number of samples queried along with the memory, the Bernoulli sampler only queries $\sim pT$ times with high probability, while the reservoir sampler queries about $pT\log T$ times. Secondly, it shows that the Bernoulli sampler *does* offer continuous robustness, but up to a flattening of the deviation control for sets of small incidence (small $W_t$). Ben-Eliezer & Yogev show a number of applications of such bounds to sketching, and Alon et al. have recently applied this to tightly characterise the regret in online classification [Alo+21], using techniques of Rakhlin et al. [RST15a; RST15b]. We believe that self-normalised bounds as above can contribute to showing adaptive versions of these results.

## B  Analysis of VUE Against Adaptive Adversaries

This section serves to show Theorems 2 and 10. We will analyse the excess abstention, and the mistakes separately. Both deviations are controlled with probability $1 - \delta/2$, and so a union bound completes the argument. The excess abstention control is common to both, and exploits Bernstein's inequality.

---

[4]In a similar vein of edge-cases, if $W_t < 1 \implies W_t = 0$, then $0 \leq \widetilde{W}_t \leq W_t = 0$, and thus the bound extends to all possible values of $W_t$.

*Proof of excess abstention bound.* Notice that the procedure only abstains if $C_t = 1$ or if $\widehat{\mathcal{Y}}_t = \{\perp\}$. In the latter case, the competitor also abstains, and thus no excess abstention is incurred. Therefore, the net excess abstention is bounded as $A_T - A_T^* \leq \sum C_t$. Now, $\sum C_t$ is a Binomial random variable with parameters $T, p$. By Bernstein's inequality [BLM13, Ch. 2],

$$\mathbb{P}\left(\sum C_t \geq pT + 2\sqrt{p(1-p)T\log(2/\delta)} + 2\log(2/\delta)\right) \leq \frac{\delta}{2}. \qquad \square$$

We move on to bounding mistakes in a $N$-dependent way.

*Proof of mistake bound from Theorem 2.* As in the main text, consider the filtration $\{\mathscr{F}_t\} = \{\sigma(\mathscr{H}_t^{\mathfrak{A}})\}$, $U_t^f := \mathbb{1}\{f(X_t) \notin \{\perp, Y_t\}\}$, and consider the processes $W_t^f = \sum_{s<t} U_t^f$, $B_t = C_t$, $\widetilde{W}_t^f = U_t^f C_t$. Note that since $N \geq 2$, $\frac{\delta}{2N} \leq \frac{1}{4} \leq \frac{1}{\sqrt{e}}$.

Note that for every $f$, $U_t^f$ and $C_t$ satisfy the requirements of Lemma 1, since $C_t$ is tossed independently of $\mathscr{H}_{t-1}^{\mathfrak{A}}$. Therefore, we may invoke Lemma 1 to find that

$$\mathbb{P}\left(\exists t : \widetilde{W}_t^f = 0, W_t^f \geq \frac{8}{p}\log(2N/\delta)\right) \leq \frac{\delta}{2N},$$

and applying a union bound over $f \in \mathcal{F}$, we conclude that

$$\mathbb{P}\left(\exists t, f : \widetilde{W}_t^f = 0, W_t^f \geq \frac{8}{p}\log(2N/\delta)\right) \leq \frac{\delta}{2},$$

Notice that if $\widetilde{W}_{t-1}^f$ is non-zero, then $f \notin \mathcal{V}_t$ since we've seen it make a mistake prior to the time $t$. Now define the stopping times $\tau_f := \max\{t : f \in \mathcal{V}_t\} = \max\{t : \widetilde{W}_{t-1}^f = 0\}$. We observe that

$$\begin{aligned}
M_T = \sum_t \mathbb{1}\{\widehat{Y}_t \notin \{\perp, Y_t\}\} &\leq \sum_t \mathbb{1}\{\exists f \in \mathcal{V}_t : f(X_t) \notin \{\perp, Y_t\}\} \\
&\leq \sum_f \sum_t \mathbb{1}\{f \in \mathcal{V}_t, f(X_t) \notin \{\perp, Y_t\}\} \\
&= \sum_f \sum_t \mathbb{1}\{t \leq \tau_f\} U_t^f.
\end{aligned}$$

Next, define the event

$$\mathsf{E} := \left\{\exists t, f : f \in \mathcal{V}_t, W_{t-1}^f \geq 8\log(2N/\delta)/p\right\}.$$

Since $f \in \mathcal{V}_t \iff \widetilde{W}_{t-1}^f = 0 \iff t \leq \tau_f$. Also recall that $W_{t-1}^f = \sum_{s<t} \mathbb{1}\{f(X_s) \notin \{\perp, Y_s\}\}$. Therefore, given $\mathsf{E}^c$,

$$\sum_t \mathbb{1}\{t \leq \tau_f, f(X_t) \notin \{\perp, Y_t\}\} \leq 8\frac{\log(2N/\delta)}{p} + 1,$$

since on $\mathsf{E}^c$, $t \leq \tau_f \implies \widetilde{W}_{t-1}^f = 0 \implies \sum_{s<t} U_t^f \leq \frac{8\log(2N/\delta)}{p}$, and the additional 1 arises since $\mathsf{E}^c$ does not control behaviour at $\tau_f$. We conclude that given $\mathsf{E}^c$, we have

$$M_T \leq \sum_f 9\frac{\log(2N/\delta)}{p} = 9\frac{N\log(2N/\delta)}{p}.$$

But $\mathsf{E}$ occurs with probability at most $\delta/2$, and we have shown that

$$\mathbb{P}\left(M_T > \frac{9N\log(2N/\delta)}{p}\right) \leq \frac{\delta}{2}. \qquad \square$$

As discussed in §5, the $\mathcal{X}$-dependent argument proceeds similarly.

*Proof of mistake bound from Theorem 10.* Again, consider the filtration $\{\mathscr{F}_t\} = \{\sigma(\mathscr{H}_t^{\mathfrak{A}})\}$. Define $\widehat{\mathcal{Y}}_t^x = \{f(x) : f \in \mathcal{V}_t\}$, and the process $U_t^x := \mathbb{1}\{X_t = x, \widehat{Y}_t \notin \{\perp, Y_t\}\}$, and consider the processes $W_t^x = \sum_{s<t} U_t^x, B_t = C_t, \widetilde{W}_t^x = U_t^x C_t$. Again, since $|\mathcal{X}| \geq 2, \frac{\delta}{2|\mathcal{X}|} \leq \frac{1}{4} \leq \frac{1}{\sqrt{e}}$.

Invoking Lemma 1, since $C_t$ is tossed independently of $\mathscr{H}_{t-1}^{\mathfrak{A}}$, we find that

$$\mathbb{P}\left(\exists t : \widetilde{W}_t^x \leq 1, W_t^x \geq \frac{8}{p}\log(2|\mathcal{X}|/\delta)\right) \leq \frac{\delta}{2|\mathcal{X}|},$$

and applying a union bound over $x \in \mathcal{X}$, we conclude that

$$\mathbb{P}\left(\exists t, x : \widetilde{W}_t^x \leq 1, W_t^x \geq \frac{8}{p}\log(2|\mathcal{X}|/\delta)\right) \leq \frac{\delta}{2},$$

Now, from the argument in the main text, $U_t^x \geq 0 \implies |\widehat{\mathcal{Y}}_t^x| \geq 2 \iff W_{t-1}^x \leq 1$. So, define the stopping times

$$\tau_x := \max\{t : |\widehat{\mathcal{Y}}_t^x| \geq 2\} = \max\{t : W_{t-1}^x \leq 1\}.$$

We have that

$$M_T = \sum_t \mathbb{1}\{\widehat{Y}_t \notin \{\perp, Y_t\}\}$$

$$= \sum_x \sum_t \mathbb{1}\{|\widehat{\mathcal{Y}}_t^x| \geq 2\}U_t^x$$

$$= \sum_x \sum_t \mathbb{1}\{t \leq \tau_x\}U_t^x.$$

Defining the event

$$\mathsf{E} := \left\{\exists t, x : t \leq \tau_x, W_{t-1}^x \geq 8\log(2|\mathcal{X}|/p)\right\},$$

we again observe that given $\mathsf{E}^c$,

$$\sum_t \mathbb{1}\{t \leq \tau_x\}U_t^x \leq 1 + 8\frac{\log(2N/\delta)}{p},$$

since on $\mathsf{E}^c$, $t \leq \tau_x \iff \widetilde{W}_{t-1}^x \leq 1 \implies \sum_{s \leq t-1} U_s^x \leq \frac{8\log(2|\mathcal{X}|/\delta)}{p}$. We thus conclude that

$$M_T \leq \sum_x 9\frac{\log(2|\mathcal{X}|/\delta)}{p} = \frac{9|\mathcal{X}|\log(2|\mathcal{X}|/\delta)}{p}.$$

But $\mathsf{E}$ occurs with probability at most $\delta/2$, and we have shown that

$$\mathbb{P}\left(M_T > \frac{9|\mathcal{X}|\log(2|\mathcal{X}|/\delta)}{p}\right) \leq \frac{\delta}{2}. \qquad \square$$

## C  Stochastic Adversaries

This section contains proofs omitted from §4, and further provides a sample-and-commit based scheme that also attains tight performance in the stochastic case.

### C.1  Performance of VUE-PROD

This section consitutes a proof of Theorem 4. We begin by controlling the excess abstentions.

*Proof of excess abstention bound.* We begin by analysing the PROD algorithm for the setting where decision sets may shrink with time. For succinctness, denote $a_t^f = \mathbb{1}\{f(X_t) = \perp\}, A_t^f := \sum_{s \leq t} a_t^f$.

**Lemma 13.** *Let $\pi_t^f$ be as in Algorithm 2. If $\eta \leq 1/2$, then for any $g \in \mathcal{V}_T$, it holds that*

$$\sum_{t,f} \pi_t^f a_t^f \leq \frac{\log N}{\eta} + A_T^g + \eta\sum_{t \leq T}(a_t^g)^2.$$

*Proof.* We follow the standard analysis of PROD, updated slightly to account for versioning. Consider the potential $W_t := \sum_{f \in \mathcal{V}_t} w_t^f$, where recall that $w_t^f = \prod_{s<t}(1 - \eta a_s^f)$. Since the weights are always non-negative, for any $g \in \mathcal{V}_T$, we have that

$$W_{T+1} \geq \prod_{t \leq T}(1 - \eta a_t^g).$$

Therefore, we have the lower bound

$$\log \frac{W_{T+1}}{W_1} \geq -\log N + \sum \log(1 - \eta a_t^g) \geq -\log N - \sum \eta a_t^g - \sum (\eta a_t^g)^2,$$

which exploits the fact that for $z \leq 1/2, \log(1 - z) \geq -z - z^2$.

To upper bound the same quantity, notice that for any $t$,

$$W_{t+1} = \sum_{f \in \mathcal{V}_{t+1}} w_{t+1}^f \leq \sum_{f \in \mathcal{V}_t} w_t^f(1 - \eta a_t^f) = W_t\left(1 - \eta \sum_f \pi_t^f a_t^f\right),$$

which again exploits that weights are non-negative, and that $\mathcal{V}_t$ is a non-increasing sequence of sets. Taking ratios and bounding $\log(1 - z)$ by $-z$, and finally summing over $t = 1 : T$, we have

$$\log \frac{W_{T+1}}{W_1} = \sum_t \log \frac{W_{t+1}}{W_t} \leq -\eta \sum_t \sum_f \pi_t^f a_t^f.$$

Rearranging the inequality obtained by sandwiching $\log \frac{W_{T+1}}{W_1}$ yields the bound. $\qquad \square$

Note that the above lemma holds generically, for any loss $\ell_t^f \leq 1$, and any sequence of shrinking decision sets. We will exploit this fact later.

For our purposes, observe that since $a_t^f$ is an indicator, $(a_t^f)^2 = a_t^f$. Thus, using Lemma 13 for $g = f^* \in \mathcal{V}_T$,

$$\sum_{t,f} \pi_t^f a_t^f \leq \frac{\log N}{\eta} + A_T^* + \eta A_T^*.$$

Now, the total abstention incurred by the learner is

$$A_T = \sum \mathbb{1}\{C_t = 1\} + \mathbb{1}\{C_t = 0, f_t(X_t) = \bot\}.$$

Exploiting the independence of the exploratory coin, we find that

$$\mathbb{E}[A_T] = pT + (1 - p)\mathbb{E}[\sum_{t,f} \pi_t^f a_t^f].$$

Invoking the above bound on $\sum_{t,f} \pi_t^f a_t^f$ and rearranging then yields that

$$\mathbb{E}[A_T] \leq pT + \frac{(1 - p)\log N}{\eta} + (1 - p)\mathbb{E}[A_T^*] + \eta(1 - p)\mathbb{E}[A_T^*].$$

Now, if $\eta = p$, then $\eta(1 - p) - p = -p^2 < 0$, and then exploiting that $A_T^* \geq 0$ yields the bound

$$\mathbb{E}[A_T - A_T^*] \leq pT + \frac{\log N}{p}. \qquad \square$$

This leaves the mistake control. The argument we present critically relies on the law $\pi_t^f$ being chosen independently of $X_t$, given $\mathcal{H}_{t-1}^{\mathcal{L}}$. This is ultimately a source of inefficiency - for instance, if $\pi_t^f$ were allowed to depend also on $X_t$, then we could enforce that non-abstaining actions are not played when $C_t = 0$, and drop the second $\log(N)/p$ term from the excess abstention bound. However, we were unable to show mistake control with only logarithmic dependence on $N$ in this situation.

*Proof of mistake bound.* The mistake control proceeds by partitioning the class $\mathcal{F}$ according to the mistake rates of individual $\mathcal{F}$s and arguing that whole groups of these are simultaneously, and quickly, eliminated from the version space without incurring too many mistakes. This fundamentally exploits the stochasticity of the setting.

To this end, define

$$\mathcal{F}_\zeta := \{f \in \mathcal{F} : 2^{-\zeta} \leq P(f(X_t) \notin \{?, Y_t\}) \leq 2^{1-\zeta}\}$$
$$\overline{\mathcal{F}}_\zeta := \{f \in \mathcal{F} : P(f(X_t) \notin \{?, Y_t\}) \leq 2^{-\zeta}\}.$$

In the following, $\zeta_0$ is a parameter for the purposes of analysis, that will be chosen later. Notice that $\mathcal{F} = \bigcup_{\zeta \leq \zeta_0} \mathcal{F}_\zeta \cup \overline{\mathcal{F}}_{\zeta_0}$.

We'll argue that all $f \in \mathcal{F}_\zeta$ are eliminated quickly (for small $\zeta$). For this, it is useful to define the stopping times

$$\tau_\zeta := \max\{t : \exists f \in \mathcal{F}_\zeta \cap \mathcal{V}_t\}.$$

Notice that for any $f \in \mathcal{F}_\zeta$,

$$P(C_t = 1, f(X_t) \notin \{\bot, Y_t\}) \geq 2^{-\zeta} p.$$

As a consequence of this and the union bound, we have the following tail inequality.

**Lemma 14.** *For any $\delta \in (0, 1)$,*

$$\mathbb{P}\left(\exists \zeta \leq \zeta_0 : \tau_\zeta > \sigma_{\delta,\zeta_0}(\zeta)\right) \leq \delta,$$

*where*

$$\sigma_{\delta,\zeta_0}(\zeta) := \frac{2^\zeta}{p} \log(\zeta_0 N/\delta).$$

With this in hand, notice that

$$M_T = \sum_t \sum_f \mathbb{1}\{f_t = f\}\mathbb{1}\{f(X_t) \notin \{\bot, Y_t\}$$

$$= \sum_t \sum_{\zeta \leq \zeta_0} \sum_{f \in \mathcal{F}_\zeta} \mathbb{1}\{f_t = f\}\mathbb{1}\{f(X_t) \notin \{\bot, Y_t\}\} + \sum_t \sum_{f \in \overline{\mathcal{F}}_{\zeta_0}} \mathbb{1}\{f_t = f\}\mathbb{1}\{f(X_t) \notin \{\bot, Y_t\}\}.$$

Next, we observe that

$$\mathbb{E}\left[\sum_{f \in \mathcal{F}_\zeta} \mathbb{1}\{f_t = f\}\mathbb{1}\{f(X_t) \notin \{\bot, Y_t\}\bigg| \mathscr{H}_{t-1}^\mathfrak{L}\right] = \sum_{f \in \mathcal{F}_\zeta} \pi_t^f P(f(X_t) \notin \{\bot, Y_t\})$$

$$\leq 2^{1-\zeta} \pi_t(f_t \in \mathcal{F}_\zeta)$$

$$\leq 2^{1-\zeta}\mathbb{1}\{t \leq \tau_\zeta\},$$

where the first equality is because $\pi_t^f$ is predictable given $\mathscr{H}_{t-1}^\mathfrak{L}$, the second uses the definition of $\mathcal{F}_\zeta$, and the final inequality is because $\pi_t$ is a distribution that is supported on $\mathcal{V}_t$, and thus has total mass at most 1, and mass 0 when $\mathcal{F}_\zeta \cap \mathcal{V}_t = \varnothing$. In much the same way, also notice that

$$\mathbb{E}\left[\sum_{f \in \overline{\mathcal{F}}_{\zeta_0}} \mathbb{1}\{f_t = f, f(X_t) \notin \{\bot, Y_t\}\}\bigg| \mathscr{H}_{t-1}^\mathfrak{L}\right] \leq 2^{-\zeta_0}.$$

Exploiting both the linearity of expectations and the tower rule,

$$\mathbb{E}[M_T] \leq \sum_t \sum_{\zeta \leq \zeta_0} 2^{1-\zeta} P(\tau_\zeta \geq t) + 2^{-\zeta_0} T$$

$$\leq \sum_{\zeta \leq \zeta_0}\left(2^{1-\zeta} \sum_{t \leq \sigma_{\delta,\zeta_0}(\zeta)} 1 + \sum_{t > \sigma_{\delta,\zeta_0}(\zeta)} \delta\right) + 2^{-\zeta_0} T$$

$$\leq 2\zeta_0 \frac{\log(\zeta_0 N/\delta)}{p} + 2\delta T + 2^{-\zeta_0} T.$$

Now set $\zeta_0 = \lfloor \log T \rfloor, \delta = 1/T$. Since $\zeta_0 N/\delta \leq N^2 T^2$, we find that

$$\mathbb{E}[M_T] \leq 4 \frac{\log T \log(NT)}{p} + 4,$$

and finally since $p \leq 1, \frac{4}{p} \geq 4$, leading to the claimed bound (for $T \geq 3$). $\qquad\square$

## C.2 Lower Bound

*Proof of Theorem 5.* Without loss of generality, assume $f_2(x) = 1$. Recall that $f_1(x) = \bot$. We describe the two adversaries -

- $P_1^\gamma$ is supported on $\{(x, 1)\}$, so that for each time $X_t = x$, and the label $Y_t = 1$.

- $P_2^\gamma$ is supported on $\{(x, 1), (x, 2)\}$ such that for each time $X_t = x$, while the label is drawn iid from the law $Y_t = \begin{cases} 1 & \text{w.p. } 1 - \gamma \\ 2 & \text{w.p. } \gamma \end{cases}$.

Notice that against $P_1^\gamma$, the competitor is $f_2$, which attains $A_T^{(P_1^\gamma)} = 0$, while against $P_2^\gamma$, the competitor is $f_1$, which attains $A_T^{(P_2^\gamma)} = T$. Observe further that since $\gamma < 1/2$, if any learner does not play $\bot$, it is advantageous for it to play 1 and never play 2.[5] We thus lose no generality in assuming that the learner's actions lie in $\{\bot, 1\}$. Now, run two coupled versions of the learner, so that if these observe the same $Z_t$s, they produce identical actions. Feed the first of these data generated from $P_1^\gamma$, and the second of these data generated from $P_2^\gamma$.

Let $\eta_1$ be the (random) number of abstentions that the first version of the learner makes - this means that it must have played 1 $T - \eta_1$ times. Denote the number of mistakes that the second version of the learner makes as $\eta_2$. Given $\eta_1$, the second version gets exactly the same sequence as the first with probability $(1 - \gamma)^{\eta_1}$ - indeed, due to the coupling, they first abstain together, and then receive the same label with probability $1 - \gamma$. Conditioned on this, they again abstain together, and then receive the same label with probability $1 - \gamma$ and so on, $\eta_1$ times. This means that, given $\eta_1$, and the event that they get the same sequence, the second version of the learner plays $T - \eta_1$ '1' actions. Since each of these is wrong with probability $\gamma$, independently and identically,

$$\mathbb{E}[\eta_2 | \eta_1] \geq (1 - \gamma)^{\eta_1} \gamma (T - \eta_1).$$

Notice that $(1 - \gamma)^{\eta_1}$ is a convex function of $\eta_1$. Thus, $\mathbb{E}[(1 - \gamma)^{\eta_1}] \geq (1 - \gamma)^{\mathbb{E}[\eta_1]} = (1 - \gamma)^K$. Further, $\mathbb{E}[-(1 - \gamma)^{\eta_1} \eta_1] \geq \mathbb{E}[-\eta_1] = -K$, and finally, for $\gamma \leq 1/2, (1 - \gamma) \geq e^{-2\gamma}$. It follows that

$$\mathbb{E}[\eta_2] \geq (1 - \gamma)^K \gamma T - \gamma K = \gamma(e^{-2\gamma K} T - K). \qquad\square$$

While here, let us also comment that the proof of Corollary 6 is mildly incomplete, since the argument requires that $\varphi \geq 2$. If instead $\varphi < 2$, then notice that setting $\gamma = 1/2$ in the above, and using that $\mathbb{E}[\eta_2] \geq \gamma((1 - \gamma)^K T - K)$, we have $\psi \geq 2^{-\varphi} \frac{T}{2} - \frac{2}{2} \geq \frac{T}{8} - 1$, which grows linearly with $T$.

## D Analysis of MIXED-LOSS-PROD Against Adaptive Adversaries

This section provides a proof of Theorem 7, and describes an adaptive variant of the same scheme, based on a doubling trick, that serves to show Theorem 9.

*Proof of Theorem 7.* Recall that the scheme runs PROD with the loss

$$\ell_t^f := \mathbb{1}\{C_t = 1\}\mathbb{1}\{f(X_t) \notin \{\bot, Y_t\}\} + \lambda \mathbb{1}\{f(X_t) = \bot\}.$$

We first observe that repeating the proof of Lemma 13 with $a_t^f$ replaced by $\ell_t^f$ gives us that for any $g \in \mathcal{V}_T$,

$$\sum_{t,f} \pi_t^f \ell_t^f \leq \frac{\log N}{\eta} + \sum_t \ell_t^g + \eta \sum (\ell_t^g)^2. \qquad (1)$$

---

[5]More formally, given any leaner, we can create the better—in expectation—learner that abstains when the given one does, and predicts 1 when the given one plays something other than $\bot$.

Note that this relation holds given the context and label processes. For $g = f^* \in \mathcal{V}_T$, we observe that $\ell_t^{f^*} = \lambda \mathbb{1}\{f^*(X_t) = \perp\}$, since by definition $f^*$ makes no mistakes. Instantiating the above with $f^*$, and noting $\sum \mathbb{1}\{f^*(X_t) = \perp\} = A_T^*$, we conclude that

$$\sum_{t,f} \pi_t^f \ell_t^f \leq \frac{\log N}{\eta} + \lambda A_T^* + \eta \lambda^2 A_T^*. \tag{2}$$

We proceed to characterise the mistakes and abstentions that the learner makes in terms of $\sum_{t,f} \ell_t^f$. To this end, notice that

$$M_T = \sum_{t,f} \mathbb{1}\{f_t = f\} \cdot \mathbb{1}\{C_t = 0\} \cdot \mathbb{1}\{f(X_t) \notin \{\perp, Y_t\}\}.$$

As a result, integrating over the randomness of the algorithm, but not over the contexts or labels, we find that

$$\mathbb{E}[M_T] = \mathbb{E}\left[\sum_{t,f} \mathbb{E}[\mathbb{1}\{f_t = f\}\mathbb{1}\{C_t = 0\}\mathbb{1}\{f(X_t) \notin \{\perp, Y_t\}\}|\mathcal{H}_{t-1}^{\mathfrak{A}}, X_t, Y_t]\right]$$

$$= \sum_{t,f} \mathbb{E}\left[\pi_t^f (1-p)\mathbb{1}\{f(X_t) \notin \{\perp, Y_t\}\}\right].$$

But, observe that

$$\mathbb{E}[\pi_t^f \ell_t^f] = \mathbb{E}\left[\mathbb{E}[\pi_t^f C_t \mathbb{1}\{f(X_t) \notin \{\perp, Y_t\} + \lambda \pi_t^f \mathbb{1}\{f(X_t) = Y_t\}|\mathcal{H}_{t-1}^{\mathfrak{A}}]\right]$$

$$= \mathbb{E}[p\pi_t^f \mathbb{1}\{f(X_t) \notin \{\perp, Y_t\}\}] + \lambda \mathbb{E}[\pi_t^f \mathbb{1}\{f(X_t) = \perp\}].$$

Therefore,

$$\mathbb{E}[M_T] = \sum_{t,f} \mathbb{E}\left[\frac{(1-p)}{p}\left(\pi_t^f \ell_t^f - \pi_t^f \lambda \mathbb{1}\{f(X_t) = \perp\}\right)\right]. \tag{3}$$

Further, notice that

$$A_T = \sum_t \mathbb{1}\{C_t = 1\} + \sum_{t,f} \mathbb{1}\{C_t = 0\}\mathbb{1}\{f_t = f\}\mathbb{1}\{f(X_t) = \perp\},$$

and thus,

$$\mathbb{E}[A_T] = \mathbb{E}\left[pT + (1-p)\sum_{t,f} \pi_t^f \mathbb{1}\{f(X_t) = \perp\}\right].$$

Moving the negative terms in (3) to the left hand side, and exploiting the above, we find that

$$\mathbb{E}[M_T] + \frac{\lambda}{p}\mathbb{E}[A_T - pT] = \frac{1-p}{p}\mathbb{E}\left[\sum_{t,f} \pi_t^f \ell_t^f\right],$$

where we note that both the terms $\mathbb{E}[M_T]$ and $\mathbb{E}[A_T - pT]$ are non-negative.

Exploiting the inequality 2 and the above relation, we conclude that

$$\mathbb{E}[M_T] + \mathbb{E}\left[\frac{\lambda}{p}(A_T - pT)\right] \leq \mathbb{E}\left[\frac{1-p}{p}\left(\frac{\log N}{\eta} + \lambda A_T^* + \eta \lambda^2 A_T^*\right)\right]. \tag{4}$$

The required bounds are now forthcoming. Dropping the $M_T$ term in the left hand side of (4), and pushing the constants $N, \eta, p, \lambda$ through the expectations,

$$\frac{\lambda}{p}\mathbb{E}[A_T - pT] \leq \frac{(1-p)\log N}{p\eta} + \frac{(1-p)\lambda}{p}\mathbb{E}[A_T^*] + \frac{\eta(1-p)\lambda^2}{p}\mathbb{E}[A_T^*]$$

$$\iff \mathbb{E}[A_T - pT] \leq \frac{(1-p)\log N}{\eta\lambda} + (1-p)\mathbb{E}[A_T^*] + \eta\lambda(1-p)\mathbb{E}[A_T^*]$$

$$\iff \mathbb{E}[A_T - A_T^*] \leq pT + \frac{\log N}{\eta\lambda} + (\eta\lambda - p)\mathbb{E}[A_T^*].$$

Taking $\eta = 1/2, \lambda \leq p$, observe that the last term is negative (since $A_T^* \geq 0$). Thus, making these substitutions and dropping the final term gives the required excess abstention control.

In a similar way, dropping the $\mathbb{E}[A_T - pT]$ term in (4) gives

$$\mathbb{E}[M_T] \leq \frac{\log N}{p\eta} + \frac{\lambda(1 + \eta\lambda)}{p}\mathbb{E}[A_T^*].$$

The claim follows on setting $\eta = 1/2$, and observing that $\eta\lambda \leq 1$. $\qquad\square$

## D.1 Adapting Rates for small $A_T^*$

### D.1.1 Deriving the form of $\widetilde{\alpha}$

We first describe a derivation of the form of $\widetilde{\alpha}$. As noted, the relevant parametrisation is $p = T^{-u}, \lambda = T^{-(u+v)}$, for $u, v \geq 0$. This, with the bounds of the previous section gives the control

$$\mathbb{E}[M_T] \leq 2T^u \log N + T^{\alpha^* - v}$$
$$\mathbb{E}[A_T - A_T^*] \leq T^{1-u} + 2T^{u+v} \log N + T^{\alpha^* - u - v}.$$

Notice that $\alpha^* - u - v \leq 1 - u - v \leq 1 - u$, since $\alpha^* \leq 1, v \geq 0$. Thus, we have the rate bounds

$$\mu = \max(u, \alpha^* - v)$$
$$\alpha = \max(1 - u, u + v)$$

Deriving the optimal $\alpha$ attainable for a fixed $\mu$ then amounts to the following convex program

$$\min \max(1 - u, u + v)$$
$$\text{s.t. } 0 \leq u \leq \mu$$
$$\max(0, \alpha^* - \mu) \leq v$$

Notice that the objective is a non-decreasing function of $v$, so the optimal choice of the same is $(\alpha^* - \mu)_+$, the smallest value it may take. This leaves us with trying to minimise $\max(1 - u, u + (\alpha^* - \mu)_+)$ for $0 \leq u \leq \mu$. The unconstrained minimum of this function occurs at $u_0 = \frac{1-(\alpha^*-\mu)_+}{2}$, which is feasible if $\mu \geq u_0$. If on the other hand $\mu < u_0$, then the max-affine function is in the decreasing branch $1 - u$, and the optimal choice of $u$ is just $\mu$. Thus, the optimum is achieved at

$$v = (\alpha^* - \mu)_+$$
$$u = \begin{cases} \frac{1-(\alpha^*-\mu)_+}{2} & 1 - (\alpha^* - \mu)_+ \leq 2\mu \\ \mu & 1 - (\alpha^* - \mu)_+ > 2\mu \end{cases} = \frac{\min(1 - (\alpha^* - \mu)_+, 2\mu)}{2}.$$

Correspondingly, $\widetilde{\alpha}$ takes the form

$$\widetilde{\alpha}(\mu; \alpha^*) = \begin{cases} \frac{1+(\alpha^*-\mu)_+}{2} & 1 - (\alpha^* - \mu)_+ \leq 2\mu \\ \max(1 - \mu, \mu + (\alpha^* - \mu)_+) & 1 - (\alpha^* - \mu)_+ > 2\mu \end{cases}.$$

But,

$$1 - (\alpha^* - \mu)_+ > 2\mu \iff 1 - \mu \geq \mu + (\alpha^* - \mu)_+,$$

and therefore

$$\widetilde{\alpha}(\mu; \alpha^*) = \begin{cases} \frac{1+(\alpha^*-\mu)_+}{2} & 1 - (\alpha^* - \mu)_+ \leq 2\mu \\ 1 - \mu & 1 - (\alpha^* - \mu)_+ > 2\mu \end{cases} = \max\left(1 - \mu, \frac{1 + (\alpha^* - \mu)_+}{2}\right).$$

### D.1.2 Adaptive Scheme and Proofs

We start by recalling the definition of $B_t^*$

$$B_t^* = \min_{f \in \mathcal{V}_t} \sum_{s \leq t} \mathbb{1}\{f(X_t) = \perp\}.$$

We will also use the term

$$\beta_t^* := \frac{\log B_t^*}{\log T}.$$

For the remainder of this section, let $\kappa := \frac{\lambda}{p}$. Recall that the optimal behaviour is attained by setting $p = T^{-u}, \kappa = T^{-v}$, where

$$u = \frac{\min(1 - (\alpha^* - \mu)_+, 2\mu)}{2}$$

$$v = (\alpha^* - \mu)_+.$$

Algorithm 4 essentially consitutes a doubling trick by setting $p$ and $\kappa$ in phases, which are indexed by non-negative integers, $n$. The scheme is parametrised by a scale parameter, $\theta$.

- We begin in the zeroth phase, with $\kappa = 1, p = T^{-\min(1,2\mu)/2}$ This phase ends when $\beta^*$ first exceeds $\mu$, at which point the first phase begins.
- At the beginning of each phase, we re-initialise the scheme.
- For $n \geq 1$, the $n$th phase ends when (the reinitialised) $\beta^*$ first exceeds $\mu + n\theta$.
- Each time the $n$th phase ends, we restart the scheme, with $\kappa = T^{-(n+1)\theta}, p = T^{-\min(1-(n+1)\theta,2\mu)/2}$.

Since the scheme is restarted in each phase, we may analyse each phase separately. Note that if $A_T \leq T^{\alpha^*}$ almost surely, then the index of the largest phase is at most $n^* = \lfloor \frac{(\alpha^* - \mu)_+}{\theta} \rfloor$ phases, since $\beta_t^* \leq \alpha^*$ always. For convenience, we set $T_n$ to be the length of the $n$th phase. Times $t_n$ correspond to rounds within the $n$th phase, and $M_{T_n}^n, A_{T_n}^n$ are the number of mistakes and abstentions incurred by the learner in the $n$th phase, while , $A_{T_n}^{*,n}$ is the number of abstentions incurred by $f^*$ in the $n$th phase.

Consider the behaviour in the $n$th phase. Let $g_n$ be the function that minimises $\sum_{s_n \leq T_n} \mathbb{1}\{g(X_t) = \perp\}$, subject to $\sum_{s_n \leq T_n} C_t \mathbb{1}\{g(X_t) \notin \{\perp, Y_t\}\} = 0$, and set the value of this optimum to $B_{T_n}^{*,n}$ By exploiting inequality (1) instantiated with $g_n$, and setting $\eta = 1/2$, we may infer that

$$\sum_{t_n \leq T_n} \pi_{t_n}^f \ell_{t_n}^f \leq 2\log N + p_n \kappa_n B_{T_n}^{*,n} + \frac{p_n^2 \kappa_n^2}{2} B_{T_n}^{*,n}.$$

As a result, reiterating the previous analysis over the $n$th phase, the number of mistakes and abstentions incurred in this phase

$$\mathbb{E}[M_{T_n}^n] \leq \frac{2\log N}{p_n} + 2\mathbb{E}[\kappa_n B_{T_n}^{*,n}]$$

$$\mathbb{E}[A_{T_n}^n - B_{T_n}^{*,n}] \leq \mathbb{E}[p_n T_n + 2\frac{\log N}{\kappa_n p_n}]$$

Further, notice that in each phase, $B_{T_n}^{*,n} \leq T^{\mu+(n+1)\theta}$, $\kappa_n = T^{-n\theta}, p_n = T^{-\min(1-n\theta,2\mu)/2}$. Substituting these into the above bounds, we have

$$\mathbb{E}[M_{T_n}^n] \leq 2T^{\min(1-n\theta,2\mu)/2}\log N + 2T^{\mu+\theta} \leq 4T^{\mu+\theta}\log N$$

$$\mathbb{E}[A_{T_n}^n - B_{T_n}^{*,n}] \leq T^{-\min(1-n\theta,2\mu)/2}\mathbb{E}[T_n] + T^{n\theta+\min(1-n\theta,2\mu)/2}\log N$$

But then, summing over the phases,

$$\mathbb{E}[M_T] = \sum_{0 \leq n \leq n^*} \mathbb{E}[M_{T_n}^n]$$

$$\leq 4T^\mu \log N \cdot (n^* + 1)T^\theta$$

$$\leq 4T^\mu \log N \cdot \frac{T^\theta}{\theta}.$$

Further,

$$\mathbb{E}[A_T - A_T^*] = \mathbb{E}[\sum_{n \leq n^*} A_{T_n}^n - A_{T_n}^{*,n}]$$

$$\leq \mathbb{E}[\sum_{0 \leq n \leq n^*} A_{T_n}^n - B_{T_n}^{*,n}]$$

$$\leq \mathbb{E}[\sum_{0 \leq n \leq n^*} T^{-\min(\mu,1-n\theta/2)} T_n] + \log N \sum_{0 \leq n \leq n^*} T^{n\theta+\min(1-n\theta/2,\mu)}$$

$$\leq \left( \sum_{n=0}^{n^*} T^{1-\min(\mu,1-n\theta/2)} + \sum_{n=0}^{n^*} T^{n\theta+\min(1-n\theta/2,\mu)} \right) \log N.$$

To simplify the above, let $n_0 = \lfloor \frac{1-2\mu}{\theta} \rfloor$, so that $\min(\mu, \frac{1-n\theta}{2}) = \mu$ for $n \leq n_0$. Notice that $n_0$ may be bigger or smaller than $n^*$. We can then write the bound as

$$\frac{\mathbb{E}[A_T - T_T^*]}{\log N} \leq \sum_{n=0}^{\min(n^*,n_0)} T^{1-\mu} + \sum_{n=\min(n^*,n_0)+1}^{n^*} T^{\frac{1+n\theta}{2}} + \sum_{n=0}^{\min(n^*,n_0)} T^{n\theta+\mu} + \sum_{n=\min(n^*,n_0)+1}^{n^*} T^{\frac{1+n\theta}{2}},$$

where we interpret $\sum_{n=i}^{j} = 0$ for $i > j$. This can further be simplified to

$$\frac{\mathbb{E}[A_T - A_T^*]}{\log N} \leq \min(n^* + 1, n_0 + 1) T^{1-\mu} + \frac{T^\mu}{T^\theta - 1} T^{(\min(n^*,n_0)+1)\theta)} + 2\mathbb{1}\{n_0 < n^*\} \frac{T^{\frac{1+(n^*+1)\theta}{2}}}{T^{\theta/2} - 1}.$$

If we further assume that $\theta$ is chosen so that $T^{\theta/2} \geq 2$, we can lower bound $T^{\theta/2} - 1 \geq T^{\theta/2}/2, T^\theta - 1 \geq T^\theta/2$ which gives the bound

$$\frac{\mathbb{E}[A_T - A_T^*]}{4 \log N} \leq (\min(n_0, n_*) + 1) \left( T^{1-\mu} + T^{\mu+\min(n_0,n^*)\theta} + \mathbb{1}\{n_0 < n^*\} T^{(1+n^*\theta)/2} \right),$$

from which we can derive the rate control

$$\alpha \leq \zeta(\mu, n_0, n^*, \theta) = \max(1 - \mu, \mu + \min(n_0, n^*)\theta, \mathbb{1}\{n_0 < n^*\}(1 + n^*\theta)/2)$$

The exact statement of the theorem is now straightforward to prove

*Proof of Theorem 9.* We run the above procedure with $\theta = \frac{2\ln 2}{\log T}$. Notice that $T^{\theta/2} \geq 2$, and that $T^\theta/\theta \leq \frac{2}{\ln 2} \log T \leq T^\varepsilon$ for large enough $T$. Therefore, mistakes are controlled at $O(T^{\mu+\varepsilon})$.

Further, for the abstention control, again $\min(n^*, n_0) + 1 \leq n_0 + 1 \leq \frac{1}{\theta} = \frac{\log T}{2\ln 2}$. Recall the abstention rate bound $\zeta$ above. It suffices to argue that $\zeta \leq \widetilde{\alpha} + \theta$, since $T^\theta = 4 = O(1)$.

To this end, first notice that

$$n_0 < n^* \iff \lfloor \frac{1-2\mu}{\theta} \rfloor < \lfloor \frac{(\alpha^* - \mu)_+}{\theta} \rfloor \implies 1 - 2\mu < (\alpha^* - \mu)_+.$$

In this case,

$$\zeta = \max\left(1 - \mu, \mu + n_0\theta, \frac{1 + n^*\theta}{2}\right)$$

$$\leq \max\left(1 - \mu, \mu + \frac{(1-2\mu)}{\theta} \cdot \theta, \frac{1 + \frac{(\alpha^* - \mu)_+}{\theta} \cdot \theta}{2}\right)$$

$$= \max\left(1 - \mu, \frac{1 + (\alpha^* - \mu)_+}{2}\right)$$

$$= \widetilde{\alpha}(\mu; \alpha^*).$$

**Algorithm 4** ADAPTIVE-MIXED-LOSS-PROD

1: **Inputs**: $\mathcal{F}$, Time $T$, Mistake rate $\mu$, Scale $\theta$.
2: **Initialise**: $n \leftarrow 0; n_{\max} \leftarrow \lceil 1/\theta \rceil; \forall f \in \mathcal{F}, w_1^f \leftarrow 1; \forall n \leq n_{\max}, \tau_n \leftarrow T$.
3: **for** $t \in [1:T]$ **do**
4:      $u \leftarrow \min(1 - n\theta, 2\mu)/2, v \leftarrow n\theta$
5:      $p \leftarrow T^{-u}, \lambda \leftarrow T^{-(u+v)}$.
6:      Sample $f_t \sim \pi_t = w_t^f / \sum w_t^f$.
7:      Toss $C_t \sim \mathrm{Bern}(p)$.
8:      **if** $C_t = 1$ **then**
9:          $\widehat{Y}_t \leftarrow \perp$
10:     **else**
11:         $\widehat{Y}_t \leftarrow f_t(X_t)$
12:     $\forall f \in \mathcal{F}$, evaluate

$$\ell_t^f = C_t \mathbb{1}\{f(X_t) \notin \{\perp, Y_t\}\} + \lambda \mathbb{1}\{f(X_t) = \perp\}$$

13:     $w_{t+1}^f \leftarrow w_t^f (1 - \eta \ell_t^f)$.
14:     Compute

$$B^* = \min_{g \in \mathcal{F}} \sum_{\tau_n < s \leq t} \mathbb{1}\{g(X_s) = \perp\}$$

$$\text{s.t.} \sum_{\tau_n < s \leq t} C_s \mathbb{1}\{g(X_s) \notin \{\perp, Y_t\}\} = 0.$$

15:     **if** $\log B^* \geq (\mu + n\theta) \log T$ **then**
16:         $n \leftarrow n + 1$
17:         $\tau_{n+1} \leftarrow t$
18:         $\forall f \in \mathcal{F}, w_{t+1}^f \leftarrow 1$.

On the other hand, if $n_0 \geq n^*$ then we have that

$$\frac{(\alpha^* - \mu)_+}{\theta} - 1 \leq \frac{(1 - 2\mu)}{\theta} \iff \mu \leq \frac{1 + \theta - (\alpha^* - \mu)_+}{2}.$$

As a result, in this case,

$$\begin{aligned}
\zeta &\leq \max\left(1 - \mu, \mu + n^*\theta\right) \\
&\leq \max\left(1 - \mu, \mu + (\alpha^* - \mu)_+\right) \\
&\leq \max\left(1 - \mu, \frac{1 + (\alpha^* - \mu)_+ + \theta}{2}\right) \\
&\leq \widetilde{\alpha}(\mu; \alpha^*) + \theta/2 \qquad\qquad\qquad\qquad\qquad\qquad\qquad \square
\end{aligned}$$

# E  Details of Experiments.

N.B. Code required to reproduce the experiments is provided at https://github.com/anilkagak2/Online-Selective-Classification.

## E.1  Dataset Details

GAS [Ver+12] dataset is a 6-way classification task based on the 16 chemical sensors data. These sensors are used to discriminate 6 gases at various levels of concentrations. The data consists of these sensor readings for over a period of 36 months divided into 10 batches. There are $13,910$ data points in this dataset. We use the first 7 batches as training set and the remaining 3 batches as test set. This split results in train and test sets with 9546 and 4364 data points respectively. The gas task contains data from 16 sensors (each of which gives 8 numbers). The standard error attained by the class we use (see below) on this is $\approx 87\%$. For the selective classification task, we use only the data from

the first 8 sensors (and thus only 64 out of 128 features). The standard error attainable for this is $\approx 67\%$. Importantly, for the GAS task, the selective classification setting we study only demands matching the performance of the best classifier with the full 16-sensor data, and thus supervision for the 8-sensor function is according to this best function. To be more concrete, denote the training data as $\{(X_i^1, X_i^2, Y_i)\}$, where $X^1$ and $X^2$ are the features from the first and second 8 sensors respectively, and $Y$ is the label. We train a classifier $g$ on this whole dataset. Then we produce the labelled dataset $\{(X_i^1, g(X_i^1, X_i^2))\}$, and train selective classifiers on this dataset. The online problem then takes the test dataset, and gives to the learner only the $X^1$ features from it. If the learner abstains, then the label $Y_t = g(X_t^1, X_t^2)$ is given to the learner.

CIFAR-10 [KH09] dataset is a popular image recognition dataset that consists of $32 \times 32$ pixels RGB images of 10 classes. It contains $50,000$ training and $10,000$ test images. We use standard data augmentations (shifting, mirroring and mean-std gaussian normalisation) for preprocessing the datasets. The best standard error attainable for this task by the models we use (see below) is $\approx 90\%$. This experiment is more straightforward to describe- selective classifiers are trained on the whole dataset. For the online problem, the test image is supplied to the learner, and if it abstains, then the true label of that image is provided as feedback.

### E.2    Training Experts

[GKS21] proposed a scheme to train classifiers with an in-built abstention option. This scheme provides a loss function, which takes a single hyper-parameter $\mu$, and is trained as a minimax program using gradient ascent-descent. The scheme then uses the outputs of this training with a second hyper-parameter $t$ to provide classification or abstention decisions. Therefore, the scheme utilises two hyper-parameters $(\mu, t)$ to control the classification accuracy and abstentions.

We trained selective classifiers using this scheme. As per their recommendation, we used 30 values of $\mu$ with 10 values equally spaced in $[0.01, 1]$ and remaining 20 values in the $[1, 16]$. For the threshold parameter $t$, we used 20 equally spaced values in $[0.2, 0.95)$. The minimax program was run with the learning rates $(10^{-4}, 10^{-6})$ for the descent and ascent respectively. Notice that the resulting set of classifiers have $20 \times 30 = 600$ functions.

Note that classification on CIFAR-10 is a relatively difficult task than GAS. Hence, we used a simpler 3-layer fully connected neural network architecture for the GAS dataset, and a Resnet32 architecture [Ide19; HZRS16] for the CIFAR-10 dataset.

### E.3    Algorithm implementation, Hyper-parameters, Compute requirements

We implemented Algorithm 5 (which relaxes the versioning in 2) using Python constructs. It has three hyper-parameters: (a) $T$ denoting the number of rounds, (b) the exploration rate $p$, and (c) $\varepsilon$ controlling the mistake tolerance. For each run, the test data points were randomly permuted, and the first $T$ of them were presented to the algorithm.

There are two main departures from the scheme in the main text. Firstly, rather than only using feedback gained when $C_t = 1$, the version space is refined whenever $\widehat{Y}_t = \bot$, allowing faster learning. Secondly, the versioning is relaxed as already described, to only exclude functions that make too many mistakes, as determined by $\varepsilon$.

An important implementation detail is that for very small $\varepsilon$, the version space may get empty before the run concludes. This is particularly relevant for small values of $\varepsilon$. As a simple fix, we modify the versioning rule so that if the version space were to become empty at the end of a round, it is not updated (and, indeed, the state of the scheme is retained, see below).

Since our experiments are CPU compute bounded, we used a machine with two Intel Xeon 2.60 GHz CPUs providing 40 cores. Both the regret-with-varying-time experiments took about 1 hour compute time, and the operating point experiments took nearly 5 hours each.

### E.4    Regret Behaviour as Time-horizon in Varied.

We use the hyperparameter $\varepsilon = 0.01$. For the sake of efficiency, we use the adaptive scheme Algorithm 6 that adapts to the time horizon, that instead varies $p$ with the number of rounds as $p_t = \min(0.1, \frac{1}{\sqrt{t}}), \eta_t = p_t$. This adaptation strategy is a standard way to handle varying horizons,

**Algorithm 5** VUE-PROD-RELAXED

1: **Inputs**: $\mathcal{F}$, Exploration rate $p$, Learning rate $\eta$, Tolerance $\varepsilon$.
2: **Initialise**: $\mathcal{V}_1 \leftarrow \mathcal{F}; \forall t, \mathcal{U}_t \leftarrow \varnothing; \forall f \in \mathcal{F}, w_1^f \leftarrow 1, o_0^f \leftarrow 0; \mathrm{Ctr}_0 \leftarrow 0.$
3: **for** $t \in [1:T]$ **do**
4:      Sample $f_t \sim \pi_t = \frac{w_t^f \mathbb{1}\{f \in \mathcal{V}_t\}}{\sum_{f \in \mathcal{V}_t} w_t^f}.$
5:      Toss $C_t \sim \mathrm{Bern}(p).$
6:      $\widehat{Y}_t \leftarrow \begin{cases} \bot & C_t = 1 \\ f_t(X_t) & C_t = 0 \end{cases}.$
7:      **if** $\widehat{Y}_t = \bot$ **then**                  ▷ Refine the version space if the exploratory coin is heads
8:          $\mathrm{Ctr}_t \leftarrow \mathrm{Ctr}_{t-1} + 1.$
9:          **for** $f \in \mathcal{V}_t$ **do**
10:              $o_t^f \leftarrow o_{t-1}^f + \mathbb{1}\{f(X_t) \notin \{\bot, Y_t\}\}$
11:              **if** $o_t^f \leq \varepsilon\mathrm{Ctr}_t + \sqrt{2\varepsilon\mathrm{Ctr}_t}$ **then**          ▷ Retain all $f$s that have error rate $< \varepsilon$ w.h.p.
12:                  $\mathcal{U}_t \leftarrow \mathcal{U}_t \cup \{f\}.$
13:          $\mathcal{V}_{t+1} = \mathcal{V}_t \cap \mathcal{U}_t.$
14:      **else**
15:          $\mathcal{V}_{t+1} \leftarrow \mathcal{V}_t.$
16:          $\forall f \in \mathcal{V}_{t+1}, o_t^f \leftarrow o_{t-1}^f$
17:          $\mathrm{Ctr}_t \leftarrow \mathrm{Ctr}_{t-1}.$
18:      **if** $\mathcal{V}_{t+1} \neq \varnothing$ **then**             ▷ Penalise Abstentions if the version space is non-empty
19:          **for** $f \in \mathcal{V}_{t+1}$ **do**
20:              $a_t^f \leftarrow \mathbb{1}\{f(X_t) = \bot\}$
21:              $w_{t+1}^f \leftarrow w_t^f \cdot (1 - \eta a_t^f).$
22:      **else**                                    ▷ $\mathcal{V}_{t+1} = \varnothing$, and so revert the state
23:          $\mathcal{V}_{t+1} \leftarrow \mathcal{V}_t.$
24:          $\mathrm{Ctr}_t \leftarrow \mathrm{Ctr}_{t-1}.$
25:          **for** $f \in \mathcal{V}_{t+1}$ **do**
26:              $o_t^f \leftarrow o_{t-1}^f.$
27:              $w_{t+1}^f \leftarrow w_t^f.$

and the observations obtained via this represent (and slightly overestimate) the regrets for when Algorithm 5 is run with $p = \eta = \frac{1}{\sqrt{T}}$. A major advantage is that this significantly increases the efficiency of the procedure, since instead of re-starting the experiment for each time horizon, we can now run for one single time horizon, and obtain representative values of regret at smaller horizons by recording the values at checkpoints corresponding to these. In the plots, we ran for $T = 4000$, and checkpointed every 250 rounds.

### E.4.1 Excess Abstention Behaviour

As noted in the main text, the excess abstention regret for both datasets is negative. This remains consistent with the theory, and likely arises since these datasets are, of course, not the worst case distributions. The excess abstentions regret are plotted below.

### E.5 Achievable Operating Points of Mistakes and Abstentions

We use Algorithm 5, instantiated with $T = 500$, and always choosing $\eta = p$. The particular values of $p, \varepsilon$ that are scanned are, as listed in the main text, 20 equally spaced values of $p$ in the range $[0.015, 0.285]$, and 10 equally spaced values of $\varepsilon$ in the range $[0.001, 0.046]$, giving in total 200 values of $(p, \varepsilon)$ pairs that are scanned over.

The post-hoc batch operating points are obtained as follows: We first find the largest value of the number of mistakes that are attained by the online learner for some choice of $(p, \varepsilon)$. Call this $M$. The values attained were $M_{\mathrm{CIFAR}} = 50$ and $M_{\mathrm{GAS}} = 120$. Then, we then instantiated the set $\mathcal{M}_{\mathrm{CIFAR}} = \{2, 3, \ldots, 50\}$, and for $\mathcal{M}_{\mathrm{GAS}} = \{2, 7, \ldots, 117\}$. The density was chosen lower for GAS

**Algorithm 6** VUE-PROD-RELAXED-TIME-ADAPTED

1: **Inputs**: $\mathcal{F}$, Tolerance $\varepsilon$.
2: **Initialise**: $\mathcal{V}_1 \leftarrow \mathcal{F}; \forall t, \mathcal{U}_t \leftarrow \varnothing; \forall f \in \mathcal{F}, w_1^f \leftarrow 1, o_0^f \leftarrow 0; \text{Ctr}_0 \leftarrow 0.$
3: **for** $t \in [1:T]$ **do**
4:      $p_t \leftarrow \min(0.1, 1/\sqrt{t}).$
5:      $\eta_t \leftarrow p_t.$
6:      Sample $f_t \sim \pi_t = \frac{w_t^f \mathbb{1}\{f \in \mathcal{V}_t\}}{\sum_{f \in \mathcal{V}_t} w_t^f}.$
7:      Toss $C_t \sim \text{Bern}(p_t).$
8:      $\widehat{Y}_t \leftarrow \begin{cases} \perp & C_t = 1 \\ f_t(X_t) & C_t = 0 \end{cases}.$
9:      **if** $\widehat{Y}_t = \perp$ **then**            $\triangleright$ Refine the version space if the exploratory coin is heads
10:          $\text{Ctr}_t \leftarrow \text{Ctr}_{t-1} + 1.$
11:          **for** $f \in \mathcal{V}_t$ **do**
12:              $o_t^f \leftarrow o_{t-1}^f + \mathbb{1}\{f(X_t) \notin \{\perp, Y_t\}$
13:              **if** $o_t^f \leq \varepsilon \text{Ctr}_t + \sqrt{2\varepsilon \text{Ctr}_t}$ **then**      $\triangleright$ Retain all $f$s that have error rate $< \varepsilon$ w.h.p.
14:                  $\mathcal{U}_t \leftarrow \mathcal{U}_t \cup \{f\}.$
15:          $\mathcal{V}_{t+1} = \mathcal{V}_t \cap \mathcal{U}_t.$
16:      **else**
17:          $\mathcal{V}_{t+1} \leftarrow \mathcal{V}_t.$
18:          $\forall f \in \mathcal{V}_{t+1}, o_t^f \leftarrow o_{t-1}^f$
19:          $\text{Ctr}_t \leftarrow \text{Ctr}_{t-1}.$
20:      **if** $\mathcal{V}_{t+1} \neq \varnothing$ **then**            $\triangleright$ Penalise Abstentions if the version space is non-empty
21:          **for** $f \in \mathcal{V}_{t+1}$ **do**
22:              $a_t^f \leftarrow \mathbb{1}\{f(X_t) = \perp\}$
23:              $w_{t+1}^f \leftarrow w_t^f \cdot (1 - \eta_t a_t^f).$
24:      **else**                         $\triangleright \mathcal{V}_{t+1} = \varnothing$, and so revert the state
25:          $\mathcal{V}_{t+1} \leftarrow \mathcal{V}_t.$
26:          $\text{Ctr}_t \leftarrow \text{Ctr}_{t-1}.$
27:          **for** $f \in \mathcal{V}_{t+1}$ **do**
28:              $o_t^f \leftarrow o_{t-1}^f.$
29:              $w_{t+1}^f \leftarrow w_t^f.$

for visual pleasantness. Finally, for each $m \in \mathcal{M}_*$, we run the post-hoc optimisation

$$a(m) := \min_{f \in \mathcal{F}} \sum_t \mathbb{1}\{f(X_t) = \perp\} \quad \text{s.t.} \quad \sum_t \mathbb{1}\{f(X_t) \notin \{\perp, Y_t\}\} \leq m.$$

The resulting points $(a(m), m)$ are plotted as black triangles.

**Definition of MMEA** As stated in the main text, the mistake matched competitor is defined as follows: suppose that the scheme makes $M$ mistakes and $A$ abstentions over a stream. If the following program is feasible, then we define

$$A^*(m) = \min_{f \in \mathcal{F}} \sum \mathbb{1}\{f(X_t) = \perp\} \text{ s.t. } \sum \mathbb{1}\{f(X_t) \notin \{\perp, Y_t\}\} \leq M.$$

If not, then we take $A^*(M)$ to be the abstentions made by the least mistake $f$, which is the competitor in the rest of the section. Then we define

$$\text{MMEA} = A - A^*(M).$$

### E.6 Sensitivity of the scheme to hyperparameters

Working in the setting of Figure 2, we show how the excess mistake and abstention regrets vary at $T = 4000$ (the final point) as $\varepsilon$ is varied in Figure 5. As expected, the excess mistakes increase roughly linearly with large $\varepsilon$, but the data reflects subtle non-monotonicities in the same. The variation in abstentions is, as expected, essentially opposite to that of the mistakes.

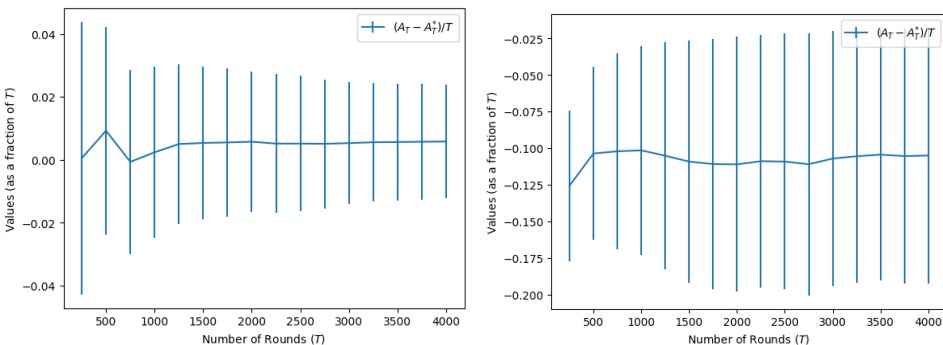

Figure 4: Excess abstention regret, normalised by $T$, in the setting of Figure 2 for CIFAR-10 (left) and GAS (right). The plots are averaged over 100 runs, and one-standard-deviation error bars are drawn. Notice that the values are negative for GAS, and strongly dominated by the MMEA for CIFAR.

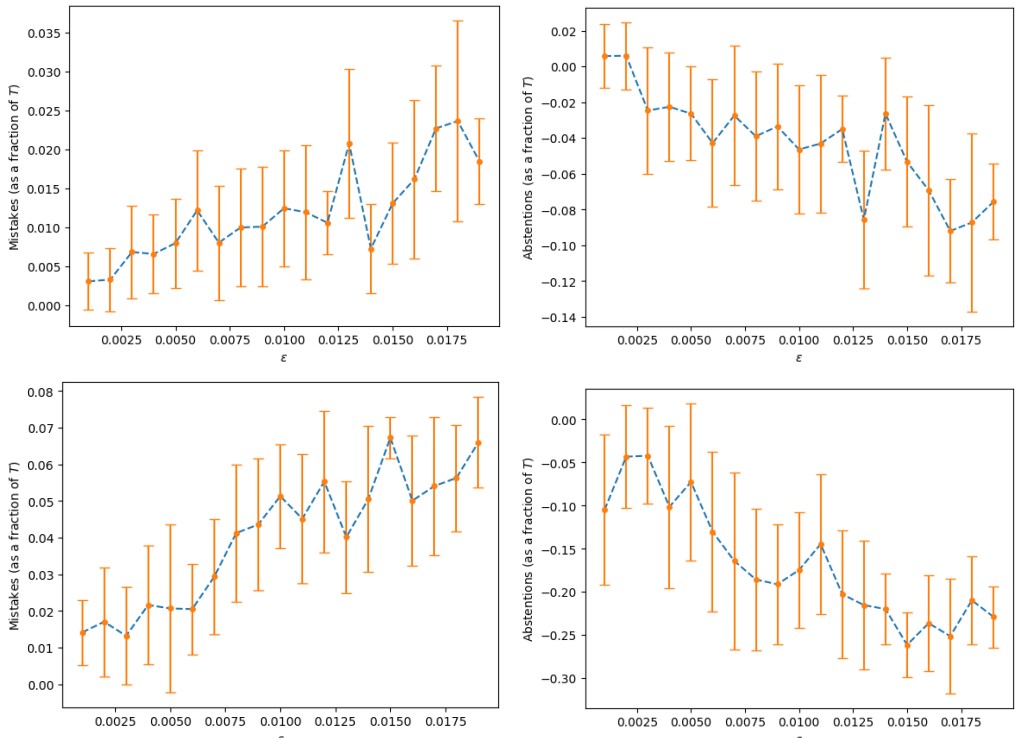

Figure 5: Senstivity with $\varepsilon$ of the excess mistakes (left) and excess abstention (right) regrets at $T = 4000$ for CIFAR (top) and GAS (bottom) datasets. Points are averaged over 100 runs, and one-standard-deviation error bars are included.

Similarly, in Figure 6, we show the operating points that can be achieved by varying $\varepsilon$ for a fixed $p$, and by varying $p$ for a fixed $\varepsilon$. We observe first that the variation with $\varepsilon$ for a fixed $p$ is relatively regular, with larger $\varepsilon$ increasing mistakes but decreasing abstentions at roughly the same rate, up to small variations. On the other, the behaviour with increasing $p$ for a fixed $\varepsilon$ is much more subtle, and indicates that a sweet-spot of the coin-based exploration rate exists for each tolerance level.

Together, these plots indicate that the optimal tuning of $\varepsilon$ and $p$ together can be subtle, and exploring how one can execute the same in an online way is an interesting open problem.

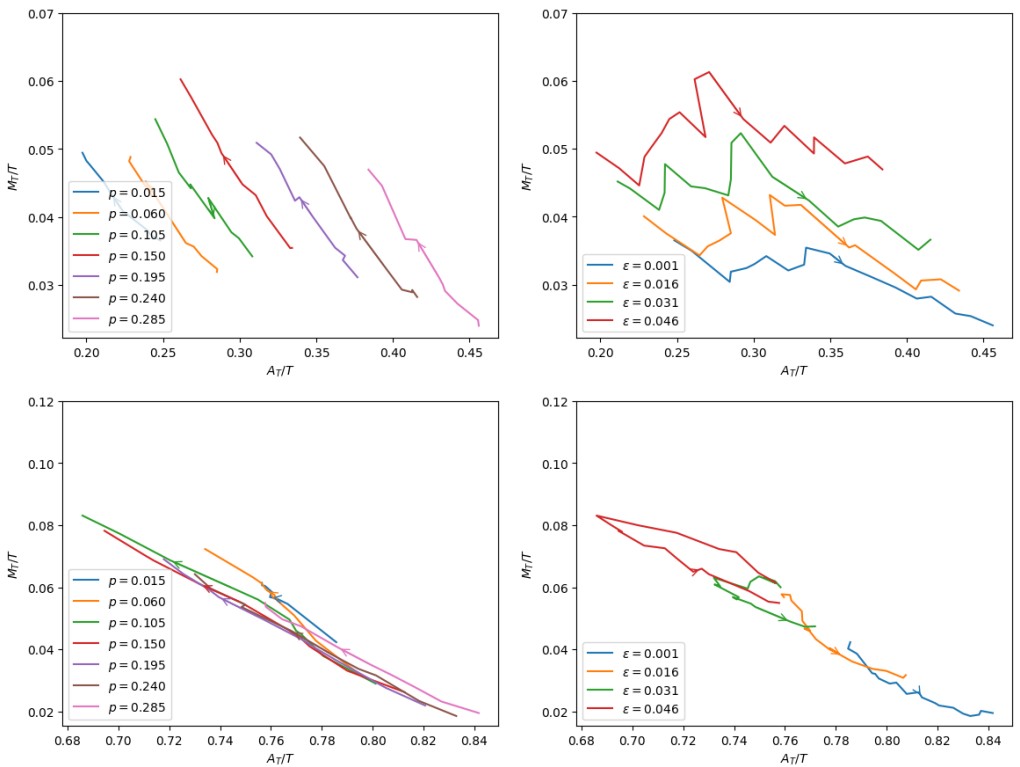

Figure 6: Illustration of how operating points achieved by the scheme vary as $p$ is changed for fixed values of $\varepsilon$ (left) and as $\varepsilon$ is changed for fixed values of $p$ (right), in the CIFAR (top) and GAS (bottom) datasets. The sets of $\varepsilon$s and $p$s marking the traces is reduced with respect to Figure 3 for the sake of legibility. The arrow denotes the direction of increasing the varied parameter.