# OpenReview forum: "Online Selective Classification with Limited Feedback"
_NeurIPS.cc/2021/Conference — NeurIPS 2021 Spotlight_

### Official Review · Reviewer_BK84 · 2021-07-16

**Rating:** 7
**Confidence:** 3

**Summary:**

The paper introduces a new setting for online classification with abstention and provide algorithms with provable guarantees for both the stochastic and adversarial setting. The bounds obtained are tight and very well exposed in the paper. The paper overall is of high quality, however, the setting studied makes multiple simplifying assumptions. Most importantly, the cost to abstention is constant, which may not be the case in real world settings. How changing this assumption affects the methods proposed is unclear. Empirical evidence serves mainly as a sanity check of the the theoretical guarantees.

**Limitations And Societal Impact:**

Yes

**Main Review:**

Originality: While online learning with abstention has been studied previously, the particulars of the observability and metrics has not been studied previously.  The algorithms provided combines different pieces from the online learning literature. The theoretical analysis is very detailed and novel on it is own. Prior work is compared to adequately.

Quality: The technical details of the paper are clearly argued for and the appendix provides full proofs for al the claims made in the paper. Empirical results do not much to the paper beyond a sanity check of the theoretical results.

Clarity: The paper is very well written and details are carefully explained. The proofs are non-trivial but are well explained to understand the technique. However, I will suggest reducing the amount of technical language in the text favor of some more basic explanations of the algorithms and what exactly do the theorems imply.

Significance: I think this paper makes a nice theoretical contribution to the study of online prediction with abstention. However, I am unsure of how their methods can be generalized to more practical scenarios. Mainly, what if there is a variable cost to abstention that depends on the context, the  setting for this is when the high capacity model may still make mistakes. Furthermore, the methods rely on the finite class assumption, I think an extension to VC classes is possible.

--------------

Post response: I have read the authors responses, although we uncovered that generalization to VC classes is harder than previously thought, the paper still remains interesting in it's own right

**Time Spent Reviewing:**

4

---

> ### Author Response · Authors · 2021-08-09
> **Response to Reviewer BK84**
>
> Thank you for your review.
>
> 1. On reducing technical language
>
>     Thanks for bringing this up. We had made an effort to present the core ideas straightforwardly, but on reflection this can be improved. Do you find the respond to Reviewer Ja1o to be in the right direction? Are there particular sections that you think we should concentrate on simplifying?
>
> 2. On extensions such as variable costs.
>
>     We note that this paper represents a first exploration at the selective classification task, and certainly while richer signalling models need to be studied for the sake of practicality, these lie beyond the scope of this study.
>
>     Of course, these questions are quite interesting, and broadly useful - e.g. along with the situation you mention, the variable cost can also arise as a pure communication cost (maybe for a mobile device that may have poor signal at times and thus suffer more latency).
>
>     Let us include some initial thoughts on this model. If the cost of abstention is known to the learner (albeit variable), then the natural strategy appears to be mixed-loss-prod, but with the $\lambda$ replaced by something proportional to this cost. The first issue to approach then seems to be to check if this is optimal, perhaps by constructing lower bounds.
>
>     Of course, this is also not the only signalling possibility here. For instance, is the variable cost observed before or after the learner's decision is made? For that matter, is it observed every time, or only when the learner abstains (and thus incurs this cost)? We will include a discussion of such extensions in a final version of the paper.
>
> 3. On extensions to VC classes.
>
>     Indeed, an extension to VC classes is possible for the stochastic case. This is quite direct via an appeal to the Sauer-shelah lemma (and a sort of lazy evaluation of the function class). Similarly Thm 7 can be generalised under finite Littlestone dimension (VC wouldn't work due to the adaptivity of the adversary), using the sequential version of Sauer-Shelah [1]. We are not so sure about Thm 2, because of the polynomial dependence on $N$, which makes naive applications of such results ineffective. In our opinion this is an interesting open problem. Again, we will include a discussion of these aspects in a full version.
>
> EDIT: We'd like to slightly qualify the above statements - as such the notion of VC classes is not directly applicable, because abstaining functions are not binary. An appropriate extension to multiclass settings should, however, suffice - for instance finite graph dimension is enough. Similarly, the sequential version would need an appropriate extension of Littlestone dimension.
>
> ----
>
> [1] Ben-David, Shai, Dávid Pál, and Shai Shalev-Shwartz. "Agnostic Online Learning." In COLT, vol. 3, p. 1. 2009.

---

> > ### Comment · Reviewer_BK84 · 2021-08-26
> > **extension to VC classes clarify**
> >
> > Can you please clarify your edit, is the extension to VC classes still possible for the stochastic case ?

---

> > > ### Author Response · Authors · 2021-08-26
> > > **Basically yes, but some care is needed.**
> > >
> > > The short answer is basically yes, the extension is possible. However, there are some technicalities that need to be handled. Below I'll describe the issue, and an standard approach that handles it.
> > >
> > > ----
> > >
> > > The technical issue arises from the fact that VC dimension itself is only defined for sets of binary valued functions. But (interesting) abstaining functions take at least 3 values ($0, 1, \bot$) so we need an appropriate modification of VC dimension in order for things to make sense.
> > >
> > > Now, notice that all for things to go through, it suffices to have some version of the Sauer-Shelah lemma, which lets us say that for finite sets of points, the number of functions involved is "effectively finite" (via a bound on the growth function). This would let us port the finite class theory (with an additional $\log T$ penalty), much as in the batch case. The question becomes if there's an appropriate notion of combinatorial dimension that gives us this for multiclass settings.
> > >
> > > One classical notion that works is the so-called graph dimension [1]. Here's a definition: if $\mathcal{F}$ is a collection of functions that take values in $[1:K],$ then for each $j \in [1:K],$ and for a function $f \in \mathcal{F},$ define $ f^j(x) = \\begin{cases} 1 & f(x) = j \\\ 0 & f(x) \neq j\\end{cases},$ which is a binary function, and construct the associated function class $\mathcal{F}^j = \\{f^j : f \in \mathcal{F}\\}.$ Then the graph dimension of $f$ is $\textrm{GD}(\mathcal{F}) = \max_j \textrm{VC}(\mathcal{F}^j).$
> > > Notice that the definition is somewhat reminiscent of one-versus-all approaches to mutliclass classification. The main point, though, is that if $\textrm{GD}$ is finite, then we can use the Sauer-Shelah lemma for each $\mathcal{F}^j$ to control the growth function of $\mathcal{F}$ (essentially by tensorisation).
> > >
> > > For us, we can view each abstaining function as something that takes values in $K+1$ classes, namely $[1:K] \cup \\{\bot\\}.$ Then if we have a class of finite graph dimension, then using the above Sauer-Shelah type lemma gives us an effectively finite class, and the theory we worked out ports directly. The "lazy evaluation" we mentioned was arising from the fact that since we get the data in a streaming way, we don't a priori know which finite set of patterns we will ultimately end up with, but this doesn't matter because at each time we can remember all the contexts that have arrived thus far, and build up the set of patterns relevant at each time on the fly.
> > >
> > > It's worth noting that graph dimension is not the only notion of relevance to multiclass classification ([1] lists others, and the Natarajan dimension also provides a bound on the growth function [2]). There's in fact further subtlety in that abstention is a different type of action than the others, so a notion like graph dimension which treats all labels the same way might not be the most efficient way to deal with it. We think these types of issues may be interesting explorations for future work.
> > >
> > > [1] Shai Ben-David and Nicolo Cesa-Bianchi and David Haussler and Philip M. Long (1995). Characterizations of Learnability for Classes of {0, ..., n}-Valued Functions. J. Comput. Syst. Sci., 50(1), 74–86.
> > >
> > > [2] Amit Daniely and Sivan Sabato and Shai Ben-David and Shai Shalev-Shwartz (2015). Multiclass learnability and the ERM principle. J. Mach. Learn. Res., 16, 2377–2404.

---

> > > > ### Comment · Reviewer_BK84 · 2021-08-30
> > > > **okay, why not instead cast the problem into the contextual bandit setting?**
> > > >
> > > > Thank you for your response. I think now things are much clearer for me.
> > > >
> > > > Can you comment a bit on why one would or would not just cast this problem as contextual bandits and use bandit algorithms?

---

> > > > > ### Author Response · Authors · 2021-08-31
> > > > > **Contextual bandit setup, as we undertand it, doesn't match our problem well. In particular, our multi-objective notions of regret and the fact that we are not allowed to observe loss when we predict fundamentally alters the problem.**
> > > > >
> > > > > In a contextual bandit problem, the operative word is bandit information, namely, for each decision the player **always** receives feedback. That is, after every round, the player predicts the output and receives feedback as to whether the output predicted is correct.
> > > > >
> > > > > In contrast, in the selective classification setting, the player can receive feedback, only when he/she chooses an abstention decision, and no feedback when output is predicted; and as such the two problems are fundamentally different.
> > > > >
> > > > > Furthermore, the interesting aspect of our paper is the **multi-objective** regret: an abstention regret that enforces how many times the player exceeds the competitor in asking for supervision labels; and a cumulative prediction regret, namely how many times additional mistakes are made by the player in comparison to the competitor.
> > > > >
> > > > > Observe that if abstention regret is not considered, namely, if we are allowed to abstain all the time, then we do receive feedback all the time, and the problem is related to a contextual bandit problem. Nevertheless, our multi-objective setting fundamentally transforms the problem. When we choose to predict, we incur (mistake) loss, but this loss is not observed by us. When we choose to abstain we incur abstention loss but do not incur a prediction loss.
> > > > >
> > > > > Thus our fundamental question is as follows: *under the condition that we do not substantially exceed the competitor with regards to the number of times we seek feedback (i.e. observe the loss), how can we also simultaneously ensure that we do not substantially exceed (unobserved) mistakes with respect to an omniscient competitor who observes all the losses?*
> > > > >
> > > > > **A side-note about motivation**
> > > > >
> > > > > See lines 17-34. Our problem is an online version of selective classification (SC).  The premise of SC is that there are easy and hard instances, and the local classifier, due to capacity constraints, even when fully trained must (occasionally) send hard instances to the cloud server. A corollary of this scenario is that the **competitor** in our online setting makes both prediction and abstention decisions. Our online problem is to compete against such a competitor on both prediction and abstention decisions, while being unable to observe losses on our prediction decisions.
> > > > >
> > > > > It is possible we missed something. We would appreciate elaboration if our comments do not satisfactorily address the reviewer's comment.

---

> > > > > > ### Comment · Reviewer_BK84 · 2021-08-31
> > > > > > **thank you!**
> > > > > >
> > > > > > Thank you for your detailed response which helped answer one of my concerns.
> > > > > >
> > > > > > I do still advise you for further work to switch to a more general framework that will enable to generalize to VC classes and variable costs more easily, however, with the current version of this work I still recommend acceptance.

---

### Official Review · Reviewer_DiTU · 2021-07-16

**Rating:** 7
**Confidence:** 4

**Summary:**

This manuscript proposes a framework for selective classification in an online fashion. In particular, it proposes strategies that make few mistakes while not abstaining too many times. A tight regret analysis is well established and experiments on real datasets demonstrate the power of the approach.

**Ethics Review Area:**

["I don’t know"]

**Limitations And Societal Impact:**

Some practical guidelines on the choice of epsilon and p should be provided.

**Main Review:**

The framework appears to be novel and the proposed strategies are impressive, supported by a tight regret analysis.

The experiments on real datasets also support the usefulness of the framework.

The paper is well written and is easy to follow. I personally enjoy reading this manuscript.

**Time Spent Reviewing:**

3

---

> ### Author Response · Authors · 2021-08-09
> **Response to Reviewer DiTU**
>
> Thank you for your positive remarks, we are glad that you enjoyed reading the paper.
>
> Regarding the choices of $\varepsilon$ and $p$,
>
> 1. For $p$ the driving design goal is how tolerant a user is to mistakes - i.e., at what mistake rate $\mu$ they're willing to operate. $p$ can be set directly in terms of this.
>
> 2. $\varepsilon$ is more subtle. There are two possibilities, as follows. We would like to note, though, that both of these are beyond the scope of the present work, and more study is needed in either situation.
>
>     1. One might introduce $\varepsilon$ as a relaxation of the underlying problem, so that the target changes to competing with the $f$ that abstains the least while making fewer than $\varepsilon T$ mistakes. This would make $\varepsilon$ into an input parameter, much like $p$.
>
>     2. Instead, one can retain the goal of competing with the classifier that makes the fewest mistakes, and use $\varepsilon$ as an internal parameter - this was the situation we looked at in section 4. The tradeoff here is between becoming too tolerant, in which case many mistake-prone functions will survive in the version space, thus raising errors, versus becoming too conservative, in which case good functions may be incorrectly thrown out, thus raising abstentions. Ultimately the correct choice of $\varepsilon$ seems to be determined by the minimum mistake rate itself. This suggests that an adaptive way to set this is needed, which can adjust $\varepsilon$ in a per-trajectory way.
>
>     In our opinion determining concrete strategies for the above constitute interesting directions for further work, and we will include a discussion of this aspect in a full version of the document.

---

### Official Review · Reviewer_Ja1o · 2021-07-16

**Rating:** 7
**Confidence:** 3

**Summary:**

This paper focuses on selective classification models in the online setting where the true label is revealed to the learner only when the abstention option is picked. Both adversarial and stochastic cases are studied in this paper, and algorithms are provided for these cases, respectively.

**Limitations And Societal Impact:**

The limitations are discussed in Section 3.
This is primarily a theoretical paper. There is no negative societal impact.

**Main Review:**

Different from the existing paper [Cor+18], this paper is focused on the setting where the true label is revealed to the learner only when the abstention option is picked, which the authors have provided real-world scenarios. Different algorithms are proposed for adversarial and stochastic cases, and detailed regret analysis are also provided for them. This paper looks quite interesting to me. But I have a few questions below. I would really appreciate it if the authors could further clarify them.
- The goal is to design an online algorithm that doesn’t abstain more times than the best-in-hindsight error-free classifier f*. However, the regret bound from Theorem 4 and 7 has a linear dependence on T, i.e., the term pT if p is a constant. Normally we would hope to get a sublinear dependence on T for the purpose of convergence. Therefore, we need to have varying p. It would be more clear if the authors could explicitly provide the formula for p in the theorems.
- It would be great if the authors could provide more explanations about the implication of Lemma 1 for regret analysis, especially in the context of Algorithm 1. In this way, it's easier to understand the rationality of how Theorem 2 is derived.
- It looks like there is no Conclusion section at the end of this paper.


**Time Spent Reviewing:**

6

---

> ### Author Response · Authors · 2021-08-09
> **Response to Reviewer Ja1o**
>
> Thanks for your review. Let us launch into the questions you ask:
>
> 1. On setting $p$
>
>     $p$ is a tunable parameter of the scheme that trades off the mistakes and abstention regrets. How to choose it ultimately depends on how these two criteria are weighted together. In the simplest case where these are weighted similarly, this can be attained by matching the regret bounds of the two cases - so, example, in Thm 4, we may pick $p = 1/\sqrt{T}$ to simultaneously get $\widetilde{O}(\sqrt{T})$ mistakes and abstentions. Importantly, observe that both the mistakes and abstention regret grow sublinearly.
>
>     Indeed, this simultaneously sublinear growth of mistakes and abstention regret was one of the major goals of this study, and in the setting of Thms 4,7, we illustrated this in Figure 1, and in corollary 8. We will explicitly include the appropriate settings of $(p,\lambda)$ needed for these. Concretely, these are as follows
>
>     1. In the setting of Thm. 4, by setting $p = T^{-\mu},$ we can control $\mathbb{E}[M_T] \le \widetilde{O}(T^\mu),$ and $\mathbb{E}[A_T - A_T^*] \le \widetilde{O}( T^\{\max(\mu, 1-\mu)\}).$
>
>     2. In the setting of Thm. 7, by setting $p = T^{-\mu/2}$, and $\lambda = T^{\mu/2 - 1}$, we can control $\mathbb{E}[M_T] \le \widetilde{O}(T^\mu)$ and $\mathbb{E}[A_T - A_T^*] \le \widetilde{O}( T^{1-\mu/2} ).$
>
>     In both cases, for a fixed $\mu,$ the abstention regrets are the tightest control possible via these schemes.
>
> 2. On Exposition of the use of Lemma 1
>
>     Thanks for this bringing this up. We had tried to express this in lines 179 - 186, but perhaps this gets a little lost. Is the following cleaner?
>
>     Ultimately the point of Lemma 1 is to let us argue that no matter what the adversary does, if we uniformly abstain at a rate $p$, then we will 'catch' any mistake prone function before it makes $O(1/p)$ mistakes. This means that with high probability, functions will fall out of the version space before they've made $O(1/p)$ mistakes.
>
>     Since the label produced by algorithm 1 must be $f(X_t)$ for _some_ $f$ in the version space, this means that the number of mistakes the learner makes is at most the number of times _any_ function in the version space is wrong. Now, by the the above argument, functions in the version space cannot ever have made more than $1/p$ mistakes, and since there are only $N$ of these, in total the number of mistakes the learner can make is at most $O(N/p).$
>
> 3. Conclusions
>
>     This was ultimately omitted due to a lack of space in the submitted version. If accepted, we would use the extra space to include a concluding discussion. Indeed, many of the points raised by the other reviewers serve as natural directions for further work and would likely feature in such a discussion.

---

> > ### Comment · Reviewer_Ja1o · 2021-09-01
> > **After reading authors' response**
> >
> > Dear authors,
> >
> > Thanks a lot for your efforts to address my concerns. I don't have other concerns about this paper as long as those points I mentioned in my original review are properly clarified in the next version. I still recommend acceptance.

---

### Official Review · Reviewer_54E7 · 2021-07-17

**Rating:** 7
**Confidence:** 3

**Summary:**

The authors propose online selective classification and explore strategies with a goal to make the least number of mistakes and abstain the fewest number of times. Concretely, they explore the adversarial case and the stochastic case. They argue that the regret bounds in the ‘versioned uniform explorer’ (VUE) scheme in the adversarial case can be improved logarithmically in N in the stochastic setting by using the VUE-PROD scheme.

**Limitations And Societal Impact:**

The authors have not discussed the limitations and societal impact of this work in as much detail.

**Main Review:**

Pros of this work:
- The introduction and related work sections are well written and easy to understand.
- The authors have described the main differences with other closely related work in a neat and concise manner.
- The authors provide strong theoretical bounds for the presented algorithms, VUE, VUE-PROD and MIXED-LOSS-PROD.


 Cons of this work:
- The experimental section in the main paper is very limited and has less details for reproducibility and is hard to understand.
- It seems that the error bars on the GAS dataset are very high for both the measures, excess mistakes and MMEA. It would be great to explain such large standard deviation error regions.

Minor typo:

L111. basis *of its history

Post Rebuttal Update: I will maintain my score of accepting this paper. I would encourage the authors to carefully go through the reviewers concerns and update their paper for the next version.

**Time Spent Reviewing:**

3-4 hours

---

> ### Author Response · Authors · 2021-08-10
> **Response to Reviewer 54E7**
>
> Thank you for your review. We are glad that you found the work easy to understand.
>
> 1. On experiments being limited.
>
>     This being a theory-focused paper, the goal of our experiment section was primarily to validate our theory, and to investigate how relaxing the competition to being slightly noisy would degrade our performance. What we found surprising, which we explained in our experiment section, is that the performance wrt abstention regret and mistake regret degrades gracefully. We think that this indicates that these ideas may have broader relevance, and serves to motivate future work.
>
>     Regarding reproducibility, we note that we have throughout justified the design choices we made in the experiments, and have provided details in appendix E. We will also release the relevant code with a final version of this paper. We hope that this addresses the issue, but would love more pointed feedback if you think this is insufficient.
>
> 2. Properties of the GAS dataset
>
>     While we are not completely clear on this, we think that ultimately this arises because the GAS task is quite challenging - for instance in the task we were studying, the best standard accuracy was $\approx 77\\%.$ We believe that the increased variance is directly a function of this, in the sense that as the scale of mistakes and MMEA increases, their variances increase with them. We would like to point out that Fig 2, GAS remains consistent with the results, since all we control in the same are the excess mistakes, which remain below $\sqrt{T\log N}$. In addition, the mean of MMEA remains below this level. We will try to probe this more closely for a full version of the paper.

---

### Decision · Program_Chairs · 2021-09-27

**Decision:**

Accept (Spotlight)

**Comment:**

This paper considers an online learning setting in which the true label is revealed only when the algorithm abstains from making a prediction. Algorithms are proposed for several variants of this setting, and strong theoretical guarantees provide a trade-off between the number of mistakes and the number of abstentions. Some experiments demonstrate that these procedures can also be applied in practice. The work advances the theoretical understanding of selective classification, and as such could be of interest to many in the NeurIPS community.